# Comparative genome analysis of *Pasteurella multocida* from Australian domestic animals suggests broad patterns of transmissions across multiple hosts and origins

Joanne L. Allen[1,2]*, Rhys N. Bushell[2], Amir H. Noormohammadi[1,2], Pam Whiteley[2,3], Susan A. Ballard[4], Mary Valcanis[4], Glenn F. Browning[1], Marc S. Marenda[1,2]

1 Asia-Pacific Centre for Animal Health, Melbourne Veterinary School, Faculty of Science, University of Melbourne, Parkville, Victoria, Australia, 2 Melbourne Veterinary School, Faculty of Science, University of Melbourne, Werribee, Victoria, Australia, 3 Wildlife Health Victoria: Surveillance, Melbourne Veterinary School, Faculty of Science, University of Melbourne, Werribee, Victoria, Australia, 4 Microbiological Diagnostic Unit Public Health Laboratory (MDU PHL), Department of Microbiology & Immunology, University of Melbourne at the Peter Doherty Institute for Infection & Immunity, Melbourne, Victoria, Australia

* jlallen@unimelb.edu.au

## Abstract

The zoonotic bacterium *Pasteurella multocida* infects a wide range of animals worldwide. While the genetic diversity of this pathogen is well described in production animals, it remains underexplored in companion animals. In Australia, most *P. multocida* genomes come from commercial poultry. Here, 59 *P. multocida* clinical isolates obtained from Australian pets, (cats, dogs, rabbits), farm animals (birds, ruminants, porcine) and captive wildlife (Quolls, Serval) between 2006 and 2023 were sequenced and compared to 523 representative RefSeq genomes. Clustering and phylogenomic analyses placed 24/25 Australian pet isolates in a long-branched clade containing several MLST profiles (ST36, ST37, ST171, ST359, ST451 and ST527) also found in human isolates. Genotypes associated with Australian production animals (e.g., ST8, ST9 and ST20 from poultry, ST79 and ST394 from cattle) fell in the main branch of the tree. Minimum spanning tree and SNP analyses suggested several occurrences of cross-species transmission. Mobile genetic elements were found across the *P. multocida* population, without clustering into any specific phylogenetic, host, or geographic group. However, a 1.8 kb cryptic plasmid (Acc. U51470), previously described in *Pasteurella canis* strains from South Korea, was detected in 99/289 (34.3%) Australian *P. multocida* isolates from various hosts, while being largely absent from the rest of the world. Antimicrobial resistance was not detected in the isolates from Australian companion animals and captive wildlife. However, resistances to tetracyclines in bovine isolates (2/10) and ampicillin in avian (1/17) isolates were identified. This study greatly expands our insights on the diversity of *P. multocida* genomes from Australian companion animals and provides the basis for wider investigations on the molecular

**Data availability statement:** All WGS files are available from GenBank (BioProject: PRJNA965909) and Illumina reads are available on the Sequence Read Archive, (Accession: SAMN40935627 – SAMN40935679, SAMN40935681 - SAMN40935686) All WGS files have been submitted to PubMLST

**Funding:** The author(s) received no specific funding for this work.

**Competing interests:** The authors have declared that no competing interests exist.

epidemiology and diversity of this pathogen, with potential applications to better understand the zoonotic risks associated with this pathogen.

## Introduction

The bacterial pathogen *Pasteurella multocida* infects production animals (poultry, pigs, large and small ruminants), companion animals (dogs, cats, rabbits), as well as horses, wildlife, and humans [1]. In animals, *P. multocida* causes septicaemic or respiratory diseases, and is occasionally isolated from soft tissue infections [2]. More rarely, the organism can be associated with zoonotic infections [3]. Whole genome sequencing (WGS) is rapidly revolutionizing our understanding of the taxonomy, virulence, host specificity, and epidemiology of this organism [4]. More than 1400 annotated *P. multocida* genomes have been deposited in the GenBank database so far, but many of these sequences come from multi-isolate projects, which often represent highly related geographic locations and/or source hosts with strong potential for duplication, leaving fewer than 500 distinct genomes in the RefSeq database [5]. While the PubMLST database [6] holds approximately 2900 genotyped isolates of *P. multocida* with diverse origins, this dataset is divided across two separate typing schemes and does not capture all publicly available genome sequences.

WGS has been recently used to explore the epidemiology of over 650 *P. multocida* [7], demonstrating that the species is composed of multiple short-branching phylogenic clades of highly related isolates, and a unique long-branching clade containing isolates from a diverse range of hosts. This illustrates the critical importance of compiling collections of isolates that offer a broad range of diversity in terms of animal source, geography and clinical history. Nevertheless, variations in common production systems (e.g., pig farms in China, rabbit farms in some European countries, cattle feedlots in USA), disease forms, and prioritized research programs may hamper the comparative analysis of datasets. Moreover, while phylogenetics and genotyping indicate a strong host predilection for certain groups of *P. multocida* strains [8], transmission of the pathogen between different hosts species is underexplored.

Due to its geographical isolation and strict quarantine regulations, Australia is free from a number of animal pathogens or has distinct pathogen populations that have evolved independently of the rest of the world. Therefore, Australian animal pathogens can be used as a model to readily examine the global epidemiology and evolution of a specific animal pathogen. The majority of Australian *P. multocida* complete genomes available in public databases (GenBank [9] or SRA [10]) were from chickens, followed by few isolates from cattle [11], humans, cats, dogs [12], and wildlife. Outside of Australia, companion animals are also underrepresented in sequence databases. Here, the genomes of 59 archival Australian *P. multocida*, including 25 isolates from clinical specimens collected from symptomatic companion animals presented at a large Veterinary Hospital, were sequenced and compared to a set of 523 genomes recently deposited in RefSeq, including 25 isolates from poultry and wild birds recently characterized by our laboratory [13].

## Materials and methods

### Culture, identification and antimicrobial susceptibility testing

The *P. multocida* strains characterized in this study were isolated from clinical specimens or necropsy samples on 5% sheep blood agar (SBA) at the Clinical Microbiology Laboratory, Melbourne Veterinary School, Werribee, Victoria, Australia. Animal ethics committee approval was not required for this study as all specimens were submitted for veterinary clinical diagnosis.

The cultures were purified and taxonomically identified at the species level by conventional biochemical tests. Stock cultures were stored at −80°C in Protect™ Multipurpose Cryobeads (Thermo Fisher Scientific). Antimicrobial susceptibility testing (AST) was performed on isolates, freshly prepared on SBA from stock cultures, by broth micro-dilution using Sensititre COMPGN1F plates (Thermo Fisher Scientific). The MIC results were read manually, and antimicrobial susceptibility was interpreted using the SWIN Expert system Version 3.3.

### Genome datasets

The major features of the isolates sequenced during this study and the published genomes are presented in S1 Table and S2 Table, respectively. The RefSeq genomes of *P. multocida* were downloaded from the NCBI Assembly database in GenBank flat file (gbff) format. The following search conditions were used: ("Pasteurella multocida"[Organism] OR Pasteurella multocida[All Fields]) AND (latest[filter] AND (all[filter] NOT anomalous[filter] AND all[filter] NOT "from large multi isolate project"[filter]) AND "refseq has annotation"[Properties]). Individual information on the strain name, subspecies, date of isolation, host and country of origin was retrieved from gbff headers. The assembled genomic sequences were individually extracted from gbff into fasta-formatted files using the EMBOSS sub-program seqret. Publicly available Illumina paired reads from Australian origin were downloaded from the Sequence Read Archive (SRA) using the prefetch and fasterq-dump tools from NCBI (https://github.com/ncbi/sra-tools).

### Whole genome sequencing

Stored *P. multocida* isolates were plated from −80°C stocks for DNA extraction and Illumina WGS by third-party laboratories, namely Australian Genome Research Facility, Parkville, Victoria, Australia (HiSeq 1000; read length up to 2 x 150 bp), Charles River Laboratories, Geelong, Victoria, Australia (NovaSeq 6000; read length 2 x 250 bp) or Microbiological Diagnostic Unit Public Health Laboratory Parkville, Victoria, Australia (NextSeq 500; read length 2 x 150 bp). Reads were processed for adapter removal and trimming of low-quality ends with TrimGalore v0.6.10 [14] using the program default settings. The genomic assembly statistics for the new genomes of this study are provided in S3 Table.

### Genomic analysis

Genomes of the *P. multocida* isolates were assembled from Illumina reads using Unicycler v0.5.0 [15]. Assembly statistics were calculated with SeqKit (subprogram stats) [16].Taxonomic identification was confirmed for each individual assembly with FastANI v1.33 [17], using a set of 94 RefSeq genomes from the *Pasteurellaceae* family (S4 Table). Assembled genomes that displayed Average Nucleotide Identity (ANI) values over 96.5% against *P. multocida* strain FDAARGOS_218 (Acc. GCF_002073255.2) were retained for further analysis. Sequence types (STs) were assessed on assembled genomes with the program mlst v2.23.0 [18], using the RIRDC and multi host schemes from the PubMLST database [6]. Prophage sequences were predicted with Phigaro v2.4.0 [19]. Plasmid sequences were predicted with PLASMe v1.1 [20] using the balance mode presets. The PLASMe output with reference results showing evidence for either a Blast hit or a transformer score superior to 0.99999 were retained for analysis. In-silico PCR was performed with ipcress (Guy St.C. Slater. <guy@ebi.ac.uk>) using the gallicida or septica primers [21] with up to 2 mismatches allowed. Antimicrobial resistance genes, as well as capsular and LPS genotypes, were interrogated with ABRicate v1.0.1 [22] using

the pre-downloaded NCBI AMRFinder database, or a bespoke database constructed from previously compiled capsular and LPS sequences [4]. Genome clustering was performed with PopPUNK v2.6.3 [23] using a Bayesian Gaussian Mixture Model. Core-genome alignments for SNP phylogeny analysis were performed on curated datasets of assembled genomes, using Parsnp [24] with the -x option for filtering out recombination. Genome datasets were simplified with the script dereplicator.py v0.3.2 (https://github.com/rrwick/Assembly-Dereplicator). Phylogenetic trees were constructed from aligned polymorphic nucleotide positions extracted from Parsnp with FastTreeDbl v2.1.11 [25] with a generalized time-reversible (GTR) model and 100 bootstraps. Trees were decorated with iToL [26]. Pairwise comparisons of numbers of SNPs (excluding complex polymorphisms) between assembled genomes were performed with snippy [27], using one sequence as reference (positional argument --ref) and the second as query (positional argument --ctgs). All-against-all genome matrix of ANI values produced by FastANI was visualized with the R package heatmap3 [28], with the "dist" function and the method "complete" for hierarchical clustering. Abundance plots were generated with the R package ggplot2 (https://ggplot2.tidyverse.org). Sequences were annotated with prokka [29] and compared with Clinker [30]. Minimum spanning trees were produced with PHYLOViZ 2.0, using the goeBURST full MST algorithm (https://doi.org/10.1093/bioinformatics/btw582).

### Sequencing data

The assembled genomes for the isolates described in this study were deposited in GenBank (BioProject: PRJNA965909). The Illumina reads are available on the Sequence Read Archive, (Accession: SAMN40935627 – SAMN40935679, SAMN40935681 - SAMN40935686). The MLST data is also available on the PubMLST website.

## Results

### Australian *Pasteurella multocida* genomes are distributed across many genomic clusters present within the general population structure

To place the latest genomic data on Australian *P. multocida* isolates in an international context, a representative sample of whole genome sequences deposited in the NCBI GenBank database (accessed 18th Sep 2024) was examined. The metadata extracted from GenBank file headers revealed the preponderance of isolates from production animals (ruminants, pigs, poultry, and to a lesser extent rabbits), predominantly originating from USA and China (Fig 1A). This trend was confirmed by the analysis of the diversity of 2933 isolates deposited in the PubMLST database, of which 1503 were placed in 29 clonal clusters, with several major ST groups isolated primarily from porcine, bovine or avian hosts (Fig 1B). In Australia, most of the publicly available whole genomic sequences from animals were related to isolates from commercial poultry farms, with only few isolates from cattle, pigs and wildlife (S4 Table). More broadly, genomic data from pets were underrepresented in public databases. Aside from 14 Australian isolates from cats (n = 12) and dogs (n = 2) recently added to GenBank, only 3 fully sequenced isolates from two dogs and a cat, originating from Greece, South Korea and the USA, were available.

A total of 59 isolates obtained from veterinary samples submitted to the MVS Clinical Diagnostic Laboratory, Werribee between 2006 and 2023 were sequenced and assembled from Illumina reads, with average coverage depth close to 300, and an average N50 of approximately 366 kbp (S3 Table). This set contained isolates from 25 pets, 29 farmed animals, one veterinary hospital environment, and 4 captive wildlife (S1 Table). These sequences were compared to 498 complete or near-complete RefSeq genomes from 27 countries downloaded from NCBI GenBank, 25 Australian avian isolates recently released by our laboratory, and a further 162 Australian genomes (148 from poultry farms in Queensland and 14 from wildlife) assembled into contigs from the paired Illumina reads retrieved from the SRA. First, the general structure of the *P. multocida* population was analyzed by calculating the core distances of the full set of 744 genomes with the program PopPUNK. Three genomes from the SRA were automatically excluded by the program, and the remaining 741 genomes were partitioned into two large groups of strains, as well as several minor clusters and singletons (Fig 2). The neighbor-joining (NJ) output trees

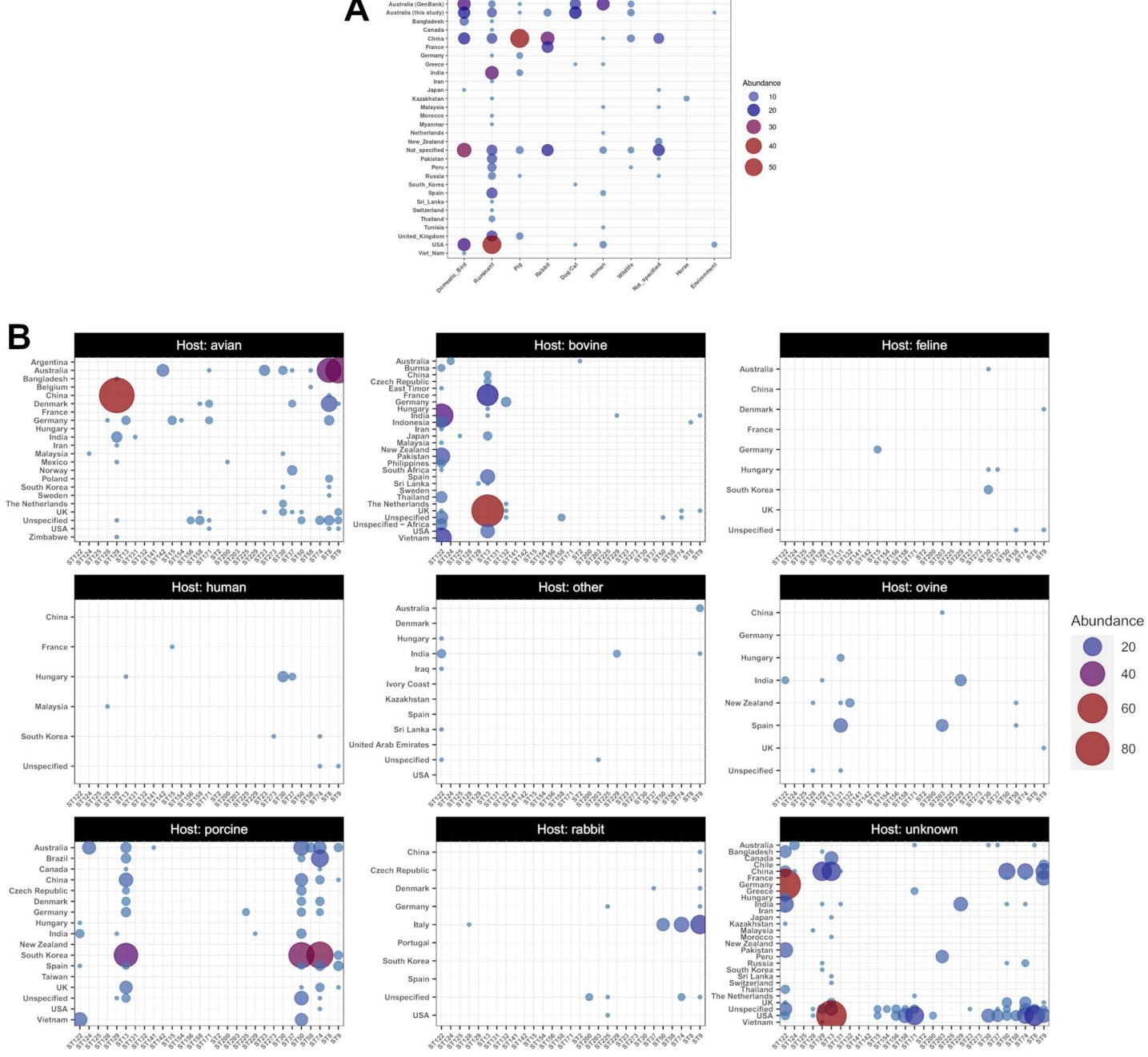

**Fig 1. Comparative abundance and origins of complete or partial *P. multocida* sequences in public databases. (A)** Country and host distribution of whole genome sequences deposited in GenBank, SRA, or characterized in this study (S1 and S2 Tables). **(B)** Country and host distribution of MLST clonal complexes from 1503 isolates genotyped by the RIRDC scheme, last updated 3rd March 2024.

from PopPUNK consistently indicated the presence of a large sub-population containing the majority of *P. multocida* genomes (Fig 2, black clade) whereas a second group formed a distinct, deep-branched cluster with a more complex structure (Fig 2, red clade). This second group contained most of the new genomes from Australian pets characterized in this study.

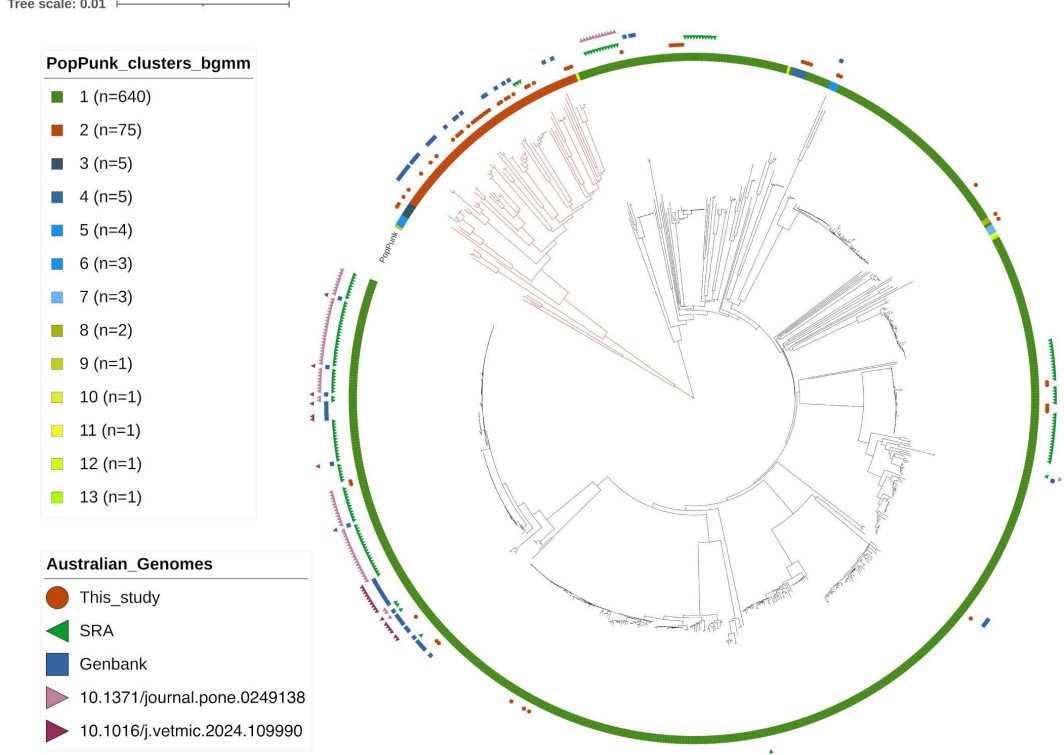

**Fig 2. Neighbor-joining tree output from the core genome distances of 741** *P. multocida* **produced by PopPUNK using a Bayesian Gaussian Mixture Model fitting.** The inner color strip indicates the core clusters assigned within the sample by the program. The sources of the Australian datasets are represented by the symbols on the outer rims, respectively indicating the current study, SRA, RefSeq/GenBank, and DOIs of recent studies [7,13].

## Australian isolates have widely diverse genotypes and fall into separate phylogenic clades

To confirm the PopPunk clustering results, a core SNP phylogenomic approach was applied to a smaller subset of *P. multocida* sequences. To simplify the analysis, the 498 complete or near-complete genomes downloaded from GenBank were first de-replicated into a smaller set of 185 distinct genomes. To build a *P. multocida* core SNP phylogeny, this set was augmented by 25 avian isolates released earlier by our laboratory, 58 genomes from humans or pets, recently deposited in RefSeq, and predominantly representing Australian isolates, 3 genomes assembled from SRA data originating from poultry isolates from Queensland, Australia and the 59 new genomes sequenced and assembled during this study. A maximum likelihood (ML) tree was constructed from the aligned polymorphic positions from these 330 genomes, after filtering out SNPs likely related to recombination. The structure of the ML tree was reminiscent of the NJ tree obtained with PopPUNK. Broadly, the mid-rooted tree was composed of two groups: a large branch containing 245 nodes and a separate deep-branching clade, containing the 85 remaining nodes (Fig 3, red clade).

The tree topology was analyzed relatively to the geographic origin of the strains, the type of hosts from which they were isolated, and the presence of specific molecular epidemiology markers (i.e., predicted capsular and LPS types, as well as MLST profiles). Full details on the complete set of genomes used in this study are listed in S1 and S2 Tables.

In the largest clade, some isolates from Australian farmed animals belonged to MLST groups described to date on this continent only (ST20 and ST30 in chickens and ST394 in cattle). Nevertheless, most of the STs from Australian production animals (e.g., ST8, ST9 and ST79) were also found in other parts of the world. In many cases, there was a noticeable

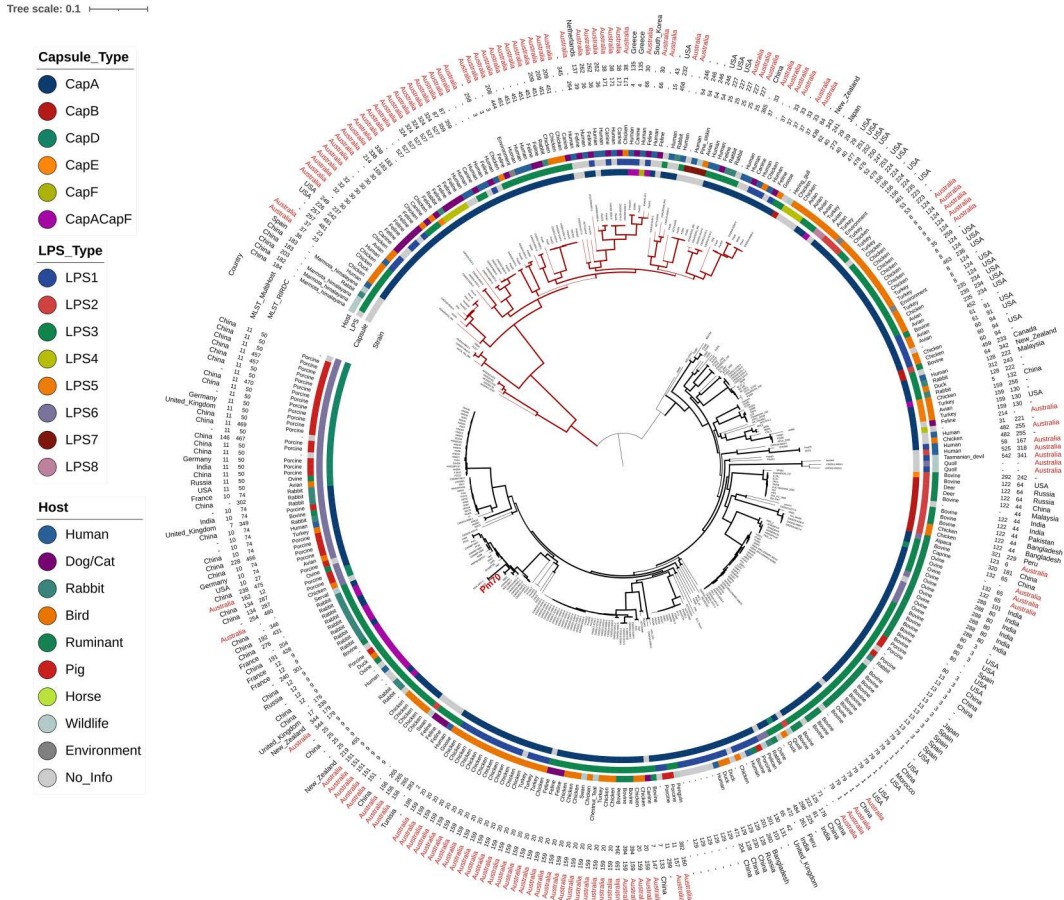

**Fig 3. Maximum likelihood phylogenetic tree from aligned SNP output from parsnp.** The tree was constructed with FastTreeDbl with 100 bootstraps and the GTR substitution model. The tree is mid-rooted, and represents 330 genomes and 77950 polymorphic positions. The color strips, starting from the outer strip, indicate the host species, LPS type and capsule type, respectively. The reference genome for *Pasteurella multocida* (PM70) is indicated in red. Branch thickness represents bootstrap values.

congruence between the MLST groups and phylogenic subclades, and to a lesser extent between STs and hosts. For example, all ST8 isolates from birds (chickens or turkeys) reared in Australia or USA were placed in the same subclade in the tree. Similarly, all ST79 isolates from cattle in Australia, USA, China and Morocco formed a compact cluster embedded in a larger and more diverse subclade, also containing ST80 (cattle isolates) and ST13 (cattle and pig isolates). Similar observations could also be made for subclades of the tree that did not contain strains from Australia. For example, all ST50 isolates were grouped into a subclade mostly composed of swine isolates from China, but contained pig isolates from India, United Kingdom and Germany, and some isolates from other hosts (a bird from USA, and a sheep from Russia) but no isolates from Australia. Likewise, the clonal complex ST74, (which also contained ST302 and ST349) formed a subclade with various hosts and countries of origin. Other MLST groups had a more complex position within the core SNP phylogeny tree, as they formed subclades with longer branch lengths and more heterogenous contents. For example, ST9 genomes were placed in a group which also contained ST301, ST204, ST176 and ST339, all belonging to the same clonal complex (ST9), as well as ST179, a genotype that differs from ST9 by 3 out of 7 alleles. Moreover, these isolates were from various countries (Australia, New-Zealand, China, France and United Kingdom) and hosts (rabbits, chickens, a duck, a sheep, and a pig).

A consistent feature of all the abovementioned subclades was the short branch lengths within each cluster, and the associations with specific, often single, MLST clonal complexes. By contrast, two other groups were particularly noticeable in the tree, as they formed long-branching nodes and contained many unassigned STs. Firstly, three isolates from Australian wildlife, including two quolls (this study) and one Tasmanian devil (GenBank acc. GCF_029852795.1) formed a small subclade more distantly related to the other genomes. Secondly, a larger cohort of 85 genomes was clearly separated from the rest of the tree (Fig 3, red clade), and contained all isolates from Australian pets described in this study, except one (CM2018-0364-0, from a cat).

Out of the 20 isolates with an unassigned ST (both with the multi-host and RIRDC schemes), 15 were Australian, primarily originating from companion animals or captive wildlife. While putative MLST or clonal complex results could be tentatively attributed to some of these genomes, several remained unclassified by either scheme (Table 1).

## Phylogeny and MLST strongly suggest multiple instances of cross-species transmission in companion animals

The deep-branching clade of 85 genomes from humans and domestic animals was extracted from the tree and examined more closely (Fig 4). It was composed of 3 main groups and organized in subclades with common STs.

Australian isolates were the most prominent within this clade, with genomes from 41 animals and 18 humans. The remaining 26 genomes were from China (forming the mostly distant subclade), Greece, Netherlands, Spain, South Korea and USA, or from unspecified countries. The branch topology and identical MLST results suggested potential occurrences of transmission between different host species in at least 5 instances within Australian isolates, requiring further analysis: (1) CM2007-0542-0 (chicken) and CM2013-0823-1 (duck), ST23; (2) CM2011-0151-1 (pet dog) and CM2006-0650-0 (farmed chicken), ST183; (3) CM2011-0584-0, CM2020-0075-0 (pet dog), CM2009-0453-0 (farmed chicken) and Past11 (human), ST171; (4) CM2014-0657-0 (pet rabbit), and CM2009-0827-0, as well as CM2019-0603-0, CM2021-0098-0, CM2018-0643-0 (pet cats), and Past26, Past31 (humans), ST527; (5) CM2013-0017-0 (pet rabbit), CM2020-1144-0, CM2021-0118-0 (pet cats), PM1447, PM1541, PM1582 (farmed chickens), Past7 and Past13 (humans), ST451.

Moreover, Australian and non-Australian strains from different hosts were also noted for the ST25 subclade, which contained two human isolates from USA (FDAARGOS_384 and FDAARGOS_385), as well as three Australian isolates from a

**Table 1. Australian isolates with unclassified STs from the Multi-host and RIRDC MLST schemes.**

| Host | Strain | Multi host (MH) MLST Scheme | | | | | | | | RIRDC MLST Scheme | | | | | | | | |
|---|---|---|---|---|---|---|---|---|---|---|---|---|---|---|---|---|---|---|
| | | Allele | | | | | | | Closest ST[3] | Allele | | | | | | | Closest ST[3] | Closest CC[3] |
| | | adk | aroA | deoD | gdhA | g6pd | mdh | pgi | | adk | est | pmi | zwf | mdh | gdh | pgi | | |
| Canine | CM2019-0177-0 | 11 | 21 | 17 | 15[2] | 18 | 17 | 18 | **41** | 16 | 20 | 33[1] | 1 | 1 | 6 | 11 | **154** | **ST154** |
| Feline | CM2010-0113-1 | 18[1] | 21 | 17 | 82 | 18 | 17 | 19 | **349** | 1[1] | 5 | 16 | 1 | 1 | 14 | 18 | **546** | – |
| Feline | CM2010-0545-2 | 19 | 21 | 32 | 16 | 18 | 57 | 37 | **114** | 1 | 27[1] | 10 | 1 | 1 | 1 | 11[1] | **15** | – |
| Feline | CM2014-0491-0 | 57 | 21 | 3 | 17 | 18 | 17 | 17 | – | 71 | 64[1] | 98 | 1 | 1 | 33 | 46 | – | – |
| Ovine | CM2019-0106-0 | 3[1] | 2 | 2 | 9 | 2 | 24 | 27 | **65** | 21[1] | 15 | 17 | 28 | 20 | 19 | 29 | **132** | **ST132** |
| Quoll | CM2011-0458-1 | 8 | 101 | 70[1] | 43[1] | 77[1] | 65 | 97 | **341** | 2 | 124[1] | 109[1] | 98[1] | 62 | 69[1] | 113 | **542** | – |
| Quoll | CM2011-0629-0 | 65[1] | 41[1] | 70[1] | 6[1] | 77[1] | 65 | 97 | **341** | 75[1] | 124 | 109[1] | 98[1] | 62 | 69[1] | 113 | **542** | – |
| Quoll | CM2019-0543-0 | 6 | 80 | 15 | 13 | 24 | 50[1] | 76[1] | – | 4 | 49[1] | 12 | 57[1] | 4 | 8 | 17[1] | – | – |
| Rabbit | CM2015-0350-0 | 19 | 26 | 18 | 16[1] | 18 | 17 | 24 | **43** | 1 | 5 | 10 | 1 | 1 | ~1[1] | 11 | **15** | **ST15** |
| Serval | CM2020-1034-0 | 5 | 45[1] | 62[1] | 8 | 68[1] | 19[1] | 34[1] | – | 6 | 67 | 2 | 60 | 57 | 3 | 116[1] | – | – |

[1]An imperfect sequence match with at least 95% identity against the allele.

[2]A sequence hit with at least 10% coverage against the allele.

[3]Determined with the PubMLST online tool.

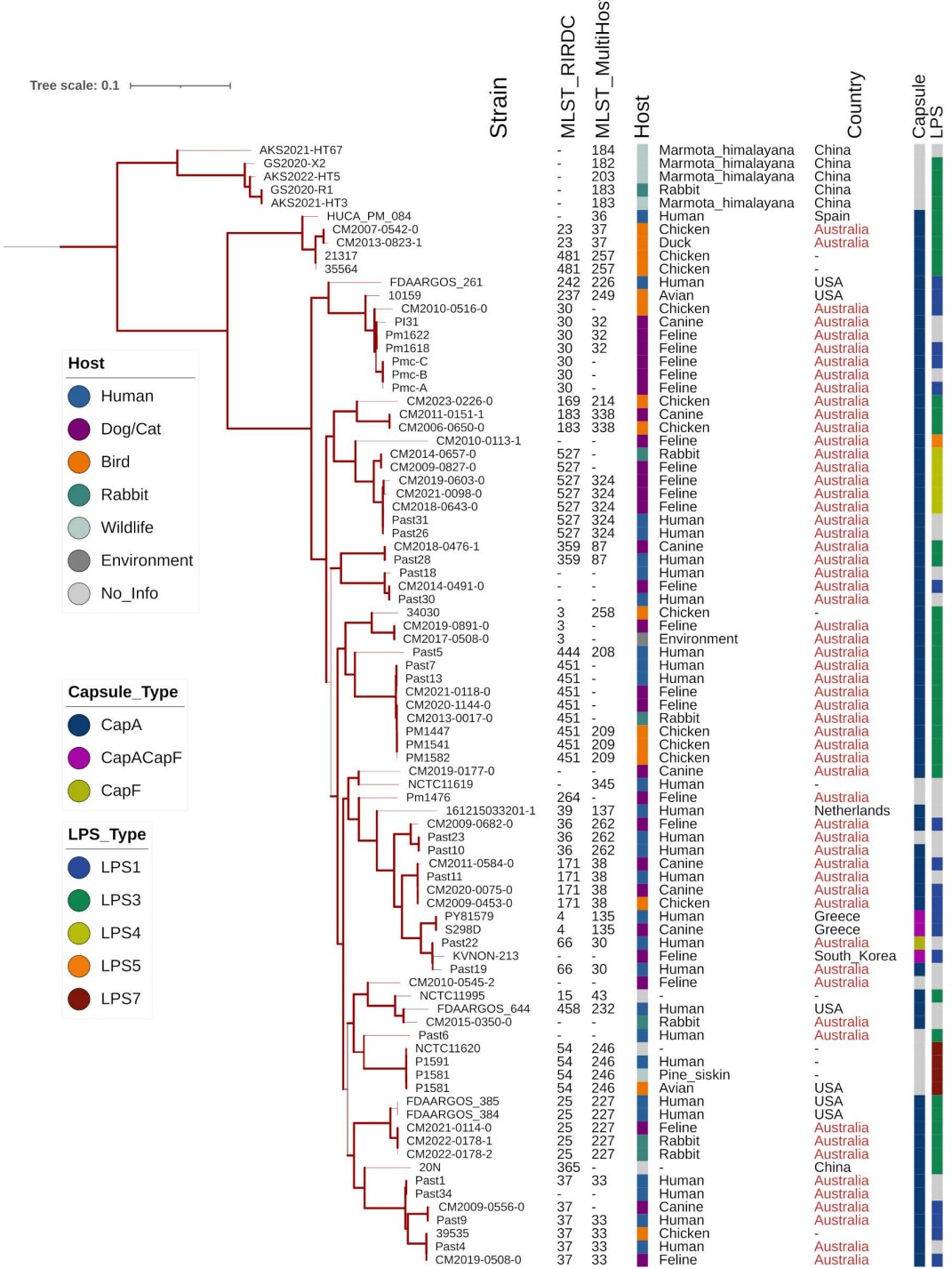

**Fig 4. Deep-branched clade extracted from the complete maximum likelihood phylogenetic tree represented in Fig 3, representing 85 genomes.** The color strips, starting from the left strip, indicate the host category, capsule type and LPS type, respectively. Branch thickness represents bootstrap values.

cat (CM2021-0114-0) and a rabbit (CM2022-0178-1 and −2, both from the same patient), on separate but closely related branches.

These observations were strengthened by inspecting a Minimum Spanning Tree computed with the PHYLOViZ tool goeBurst, using an alignment of 58913 core polymorphic nucleotide positions (each considered as a distinct locus) extracted from 585 *P. multocida* genomes. The goeBurst output had a median of 455 loci differences between isolates, and a maximum of 11775 differences between the two most distantly related genomes (Fig 5). The dataset could be sub-divided into two large groups of 505 and 80 genomes, reminiscent of the two large clades found in the ML phylogenomic tree, with the smaller group (Fig 5 Inset B) containing most of the strains of the deep-branched clade.

More occurrences of transmissions between different host species, including zoonotic, were found by progressively relaxing the maximum number of differences allowed between genomes. These were exemplified by CM2021-0118-0 (cat) and GCF_035521135.1 (human) differing only by 9 loci in ST451; CM2019-0508-0 (cat) and GCF_035521215.1 (human), differing only by 10 loci in ST37; U18-0643–0 (cat) and GCF_035520955.1 (human) differing only by 17 loci in ST527; and CM2009-0453-0 (chicken), differing only by 9 and 11 loci from CM2020-0075-0 (dog) and GCF_035521155.1 (human), respectively, in ST 171. Moreover, the ST20 genomes from chickens, cats, turkeys and wild birds identified in Table 2 (goeBURST group A) was connected to a larger set of interrelated isolates with 3–14 loci differences between distinct host species (Fig 5 Inset A).

The analysis of pairwise variations calculated by the program snippy using the same dataset of whole assembled genome sequences confirmed the close relatedness between isolates from different origins. In several cases, fewer than

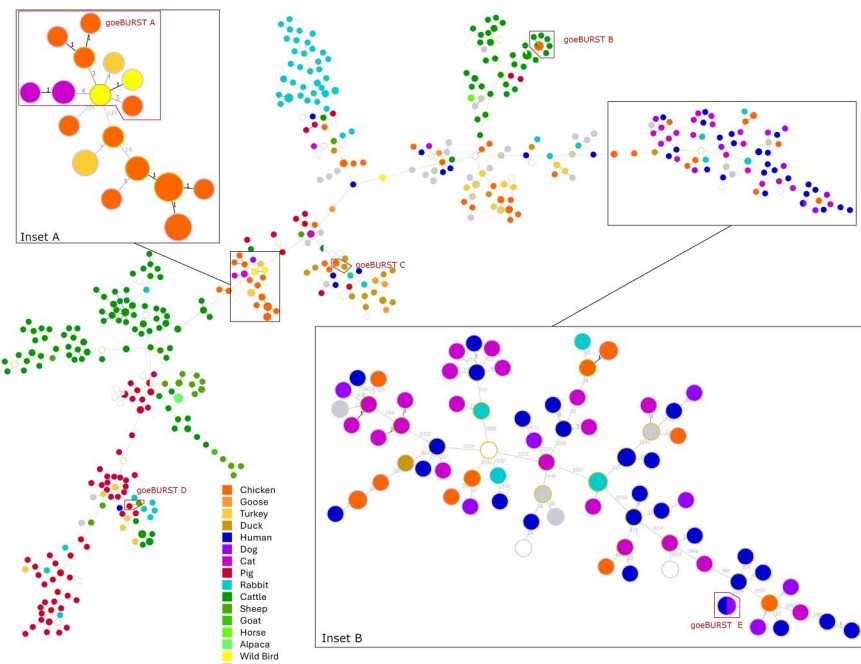

**Fig 5. Minimal spanning tree from PHYLOViZ goeBURST analysis of whole genomes of *P. multocida*.** Highlighted sections of the analysis: Inset A, branch containing ST20 isolates; Inset B, clade containing most isolates from companion animals and wildlife in the study. The distance labels in Insets A and B indicate the number of different polymorphic positions between two isolates or sets of isolates. The locations of the isolates within goeBURST A-E containing mixed host species are also indicated. A total of 76 groups of isolates with a closest genome displaying 0 to 4 loci differences, representing 525 genomes (S6 table), were considered to represent clonal complexes. Of those, five groups (goeBURST A-E) contained isolates from different host species, suggesting cross-species transmission. Within each group, the genomes had a unique ST and originated from close or identical geographical areas (Table 2).

**Table 2.** The genomes of the 5 clonal complexes containing mixed host species visualized by PHYLOViZ goeBURST analysis.

| goeBURST | MLST | Country | Host | Strain | Accession |
|---|---|---|---|---|---|
| A | 20 | Australia | Chicken | CM2017-0740-0 | CM2017-0740-0 |
| | | | | CM2017-0740-1 | CM2017-0740-1 |
| | | | | CM2017-0740-2 | CM2017-0740-2 |
| | | | Feline | Pm1612 | GCF_035521535.1 |
| | | | | Pm1613 | GCF_035520915.1 |
| | | | | Pm1620 | GCF_035520835.1 |
| | | | Chestnut_Teal | CM2013-0203-0 | CM2013-0203-0 |
| | | | Turkey | CM2022-0138-0 | CM2022-0138-0 |
| | | | Swan | CM2013-0425-0 | CM2013-0425-0 |
| B | 122 | Bangladesh | Chicken | Alim_FC_1000 | GCF_026739035.1 |
| | | | | Alim_FC_1001 | GCF_026739075.1 |
| | | | | Alim_FC_1002 | GCF_026738975.1 |
| | | | | Alim_FC_1003 | GCF_028826065.1 |
| | | | | Ban-PM4 | GCF_014338445.1 |
| | | | | Ban-PM7 | GCF_014338465.1 |
| | | | Duck | DC2020 | GCF_022575965.1 |
| | | | Bovine | BAUTB2 | GCF_003268295.1 |
| | | India | Bovine | NIVEDIPm35 | GCF_022869075.1 |
| | | | | PmBUFF2016HRY | GCF_022213185.1 |
| | | | | VTCCBAA264 | GCF_000296345.2 |
| | | Pakistan | Bovine | BUKK | GCF_001029495.1 |
| C | 129 | China | Chicken | C48-1 | GCF_001662525.1 |
| | | | | | GCF_004286945.1 |
| | | | Duck | HB02 | GCF_001661585.1 |
| D | 74 | Russia | Porcine | 1231 | GCF_029324765.1 |
| | | | Ovine | T-80 | GCF_029324775.1 |
| E | 4 | Greece | Human | PY81579 | GCF_002930775.1 |
| | | | Canine | S298D | GCF_002930755.1 |

100 SNPs were found in isolates from different hosts species, suggesting recent genomic divergences, in agreement with their MLST profiles. For example, in ST25, only 34 SNPs were detected between the feline isolate CM2021-0114-0 and rabbit isolates CM2022-0178-1 or −2; in ST171, 42–52 SNPs were present between the avian isolate CM2009-0453-0, the canine isolate CM2020-0075-0 and the human isolate GCF_035521155.1 (Past11); in ST527, 83 SNPs were present between the feline isolate CM2009-0827-0 and rabbit isolate CM2014-0657-0 (Table 3). By contrast, in each of the ST30, ST36 and ST37 sub-clades, isolates from different host species displayed approximately 1000–5000 SNPs between them, while isolates with different STs within the deep-branched clade typically had over 10000 SNPs between them.

## Mobile genetic elements are variably distributed within phylogenic groups

The SNP phylogenic analyses described above were performed on a subset of 330 *P. multocida* genomes, after de-replication of the dataset downloaded from GenBank and filtering out the polymorphic positions likely associated with recombination or accessory sequences. While this approach allowed for an unbiased representation of the phylogenetic relations between isolates in a simplified collection of *P. multocida*, it was not intended to explore the distribution of mobile genetic elements (MGEs) in the whole dataset of genomes. To address this question, sequence signatures for phages,

**Table 3. Number of SNP differences in paired *P. multocida* assembled genomes from MLST groups with putative cross-species transmissions.**

| | | | | | | | |
|---|---|---|---|---|---|---|---|
| **ST23** | CM2007-0542-0 | CM2013-0823-1 | | | | | |
| CM2007–0542–0[a] | 0 | 512 | | | | | |
| CM2013–0823–1[b] | 516 | 0 | | | | | |
| **ST25** | GCF_002393385.1 | GCF_002591295.1 | CM2021-0114-0 | CM2022-0178-1 | CM2022-0178-2 | | |
| GCF_002393385.1 | 0 | 0 | 2458 | 2208 | 2209 | | |
| GCF_002591295.1[c] | 0 | 0 | 2447 | 2210 | 2212 | | |
| CM2021–0114–0[d] | 2458 | 2447 | 0 | 2185 | 2192 | | |
| CM2022–0178–1[e] | 2208 | 2210 | 2185 | 0 | 34 | | |
| CM2022–0178–2[e] | 2209 | 2212 | 2192 | 34 | 0 | | |
| **ST30** | GCF_035221255.1 | GCF_035520775.1 | GCF_035520695.1 | GCF_035520755.1 | GCF_035520795.1 | GCF_035520655.1 | CM2010-0516-0 |
| GCF_035221255.1[d] | 0 | 10 | 879 | 870 | 869 | 1040 | 2542 |
| GCF_035520775.1[d] | 12 | 0 | 888 | 883 | 888 | 1053 | 2558 |
| GCF_035520695.1[d] | 879 | 888 | 0 | 9 | 21 | 1737 | 3243 |
| GCF_035520755.1[d] | 872 | 875 | 7 | 0 | 17 | 1736 | 3253 |
| GCF_035520795.1[d] | 884 | 890 | 20 | 17 | 0 | 1747 | 3263 |
| GCF_035520655.1[f] | 1046 | 1052 | 1722 | 1725 | 1732 | 0 | 3489 |
| CM2010–0516–0[a] | 2562 | 2572 | 3241 | 3250 | 3257 | 3502 | 0 |
| **ST36** | GCF_035521055.1 | GCF_035521175.1 | CM2009-0682-0 | | | | |
| GCF_035521055.1[c] | 0 | 407 | 2075 | | | | |
| GCF_035521175.1[c] | 401 | 0 | 2098 | | | | |
| CM2009–0682–0[d] | 2081 | 2112 | 0 | | | | |
| **ST37** | CM2019-0508-0 | GCF_035521215.1 | GCF_023556015.1 | GCF_035520895.1 | GCF_035521255.1 | GCF_035224565.1 | CM2009-0556-0 |
| CM2019–0508–0[d] | 0 | 224 | 385 | 3487 | 3087 | 4936 | 5025 |
| GCF_035521215.1[c] | 232 | 0 | 655 | 3218 | 3237 | 4872 | 5019 |
| GCF_023556015.1[a] | 386 | 644 | 0 | 3518 | 3872 | 5573 | 5506 |
| GCF_035520895.1[c] | 3496 | 3218 | 3537 | 0 | 52 | 3387 | 2976 |
| GCF_035521255.1[c] | 3093 | 3229 | 3896 | 50 | 0 | 3201 | 3019 |
| GCF_035224565.1[c] | 4914 | 4840 | 5559 | 3353 | 3165 | 0 | 547 |
| CM2009–0556–0[f] | 5011 | 5000 | 5525 | 2958 | 3008 | 547 | 0 |
| **ST171** | GCF_035521155.1 | CM2020-0075-0 | CM2009-0453-0 | CM2011-0584-0 | | | |
| GCF_035521155.1[c] | 0 | 46 | 52 | 502 | | | |
| CM2020–0075–0[f] | 46 | 0 | 42 | 502 | | | |
| CM2009–0453–0[a] | 51 | 46 | 0 | 504 | | | |
| CM2011–0584–0[a] | 503 | 499 | 506 | 0 | | | |
| **ST183** | CM2006-0650-0 | CM2011-0151-1 | | | | | |
| CM2006–0650–0[a] | 0 | 357 | | | | | |
| CM2011–0151–1[f] | 359 | 0 | | | | | |

*(Continued)*

**Table 3.** (Continued)

ST23 — ST451

| | CM2007-0542-0 SRR12130735 | CM2013-0823-1 SRR12130737 | SRR12130738 | CM2020-1144-0 | GCF_035521115.1 | CM2021-0118-0 | GCF_035521135.1 | CM2013-0017-0 |
|---|---|---|---|---|---|---|---|---|
| SRR12130735[a] | 0 | 3 | 5 | 44 | 172 | 179 | 274 | 350 |
| SRR12130737[a] | 3 | 0 | 4 | 45 | 172 | 179 | 274 | 351 |
| SRR12130738[a] | 5 | 5 | 0 | 38 | 166 | 171 | 266 | 345 |
| CM2020–1144-0[d] | 44 | 45 | 38 | 0 | 287 | 375 | 398 | 359 |
| GCF_035521115.1[c] | 177 | 177 | 169 | 292 | 0 | 227 | 172 | 604 |
| CM2021–0118-0[d] | 179 | 179 | 171 | 377 | 224 | 0 | 374 | 680 |
| GCF_035521135.1[c] | 275 | 275 | 267 | 392 | 174 | 373 | 0 | 696 |
| CM2013–0017-0[e] | 334 | 336 | 329 | 343 | 581 | 671 | 685 | 0 |

ST527

| | GCF_035520955.1 | GCF_035521015.1 | CM2018-0643-0 | CM2009-0827-0 | CM2014-0657-0 | CM2021-0098-0 | CM2019-0603-0 |
|---|---|---|---|---|---|---|---|
| GCF_035520955.1[c] | 0 | 54 | 175 | 2050 | 2062 | 954 | 1341 |
| GCF_035521015.1[c] | 54 | 0 | 201 | 1982 | 1994 | 947 | 1312 |
| CM2018–0643-0[d] | 173 | 204 | 0 | 2177 | 2190 | 1109 | 1513 |
| CM2009–0827-0[d] | 2048 | 1981 | 2178 | 0 | 83 | 2922 | 3290 |
| CM2014–0657-0[e] | 2058 | 1990 | 2186 | 83 | 0 | 2932 | 3302 |
| CM2021–0098-0[d] | 954 | 950 | 1112 | 2923 | 2938 | 0 | 2011 |
| CM2019–0603-0[d] | 1340 | 1310 | 1512 | 3293 | 3304 | 2013 | 0 |

Host species: [a]commercial poultry; [b]duck; [c]human; [d]cat; [e]rabbit; [f]dog.

plasmids, and conjugative transposons were predicted for each genome separately, and the combined results were placed on a larger phylogenic tree representing a non-dereplicated dataset (Fig 6).

There was no apparent clustering of any specific MGE sequences within any of the tree's clades. Rather, diverse repertoires of MGEs were found in most genomes, originating from a variety of countries and/or hosts, with the notable exception of a cryptic 1805 bp plasmid, NZ_U51470. This unnamed plasmid was predicted in 75 Australian isolates from different phylogenic, MLST, and LPS groups, but was absent from isolates from the rest of the world, apart from 3 strains, namely NCTC11620, P1581 and P1591, representing highly related ST54 genomes (Table 4). A BlastN search of U51470 in non-redundant nucleotide databases identified a nearly identical plasmid sequence, pHL1500, in *P. canis* strains from South Korea.

Apart from U51470, the most frequently detected plasmid sequence was related to a ∼ 325 kbp unnamed plasmid (CP020346) from *P. multocida* strain CIRMBP-0884, with a PLASMe hit observed in 399 genomes from 28 different countries.

## Subspecies classification entered in published genomes does not match phylogeny closely

A minority of *P. multocida* GenBank files examined in this study explicitly mentioned the subspecies "*multocida*", "*gallicida*" (dulcitol-positive strains), or "*septica*" (sorbitol-negative strains), in the header. In the absence of other validly

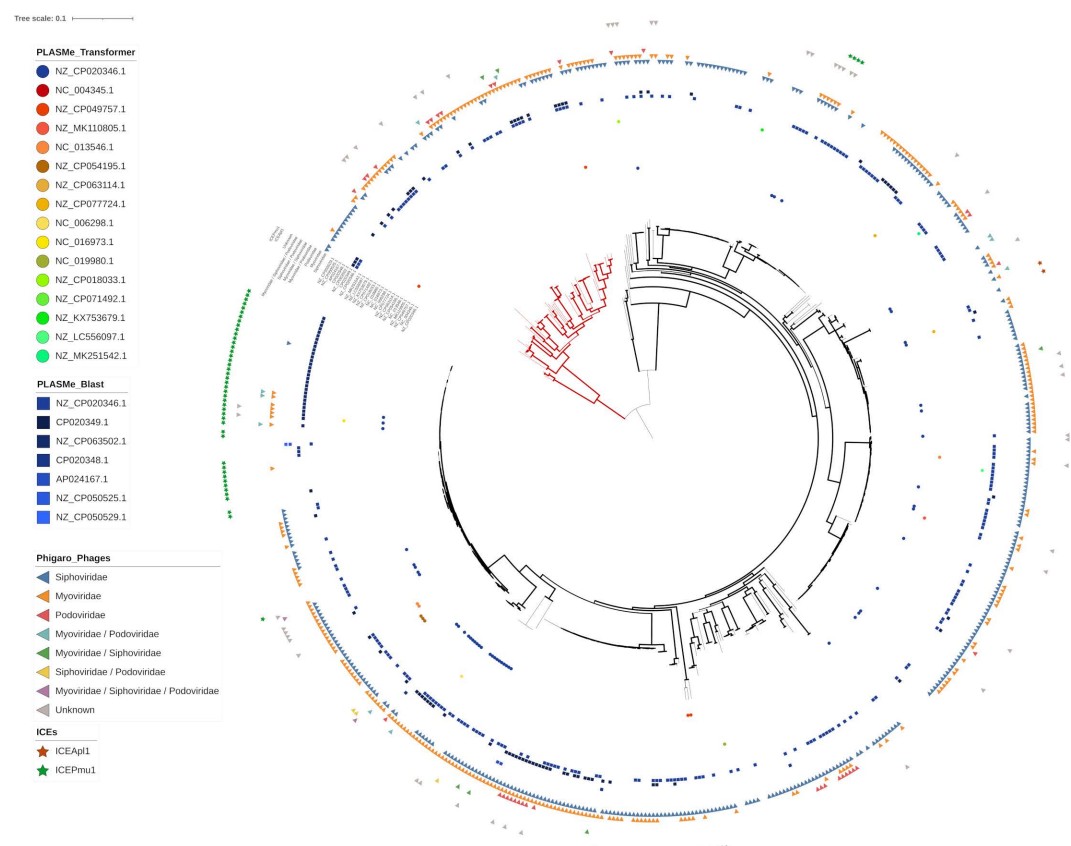

**Fig 6. Distribution of mobile genetic elements in 585 *P. multocida* genomes.** The maximum likelihood phylogenetic tree was constructed from aligned SNPs of all genomes, after correction for recombination, using FastTreeDbl with 100 bootstraps and the GTR substitution model. The tree is mid-rooted; it represents 585 genomes and 58913 polymorphic positions. Phages were predicted with Phigaro and represented by triangles. Plasmids were predicted with PLASMe and represented by circles (Transformer evidence) or squares (BlastN evidence). Conjugative transposons were predicted by BlastN searches against the IceBerg 2.0 database and represented by stars.

**Table 4. Distribution of a cryptic 1805 bp plasmid U51470 amongst *P. multocida* isolates.**

| Country | Host | LPS | RIRDC MLST | Total |
|---|---|---|---|---|
| Australia | Chicken | NA | 20 | 1 |
| | | LpsL1 | 20 | 46 |
| | | LpsL3 | 9 | 3 |
| | | | 20 | 44 |
| | | | NA | 1 |
| | Human | NA | 482 | 1 |
| | | LpsL3 | NA | 1 |
| | Rabbit | NA | NA | 1 |
| | | LpsL3 | 25 | 1 |
| Russia | Deer | LpsL2 | 122 | 1 |
| USA | Avian | LpsL7 | 54 | 1 |
| Not specified | Not specified | LpsL7 | 54 | 1 |
| | Human | LpsL7 | 54 | 1 |
| | Pine siskin | LpsL7 | 54 | 1 |
| **Total** | | | | 104 |

published subspecies epithets, the taxonomic information of most genomes beyond the species rank was unclear. To explore the genetic relatedness of these groups, the subspecies classification, when available, was retrieved from the GenBank file headers and plotted on a complete phylogenetic tree (Fig 7A), or on a matrix of ANI values constructed from pairwise comparisons between each genome (Fig 7B). The clustering analysis of ANI values divided the set into two main subgroups of genomes, which was reminiscent of the two major clades identified by PopPUNK and core SNP phylogeny (Figs 2 and 3). However, both approaches failed to cluster the *gallicida* or *septica* subspecies into a single clade or group, pointing instead to distinct genome sets (Fig 7). To verify the consistency of the subspecies classification, two pairs of PCR primers, recently proposed to differentiate *gallicida* and *septica*, were tested *in-silico* for their potential to amplify the expected products, using the genome sequences explored in this study as templates. With 2 allowed mismatches per primer, this virtual PCR assay confirmed most *gallicida*, as well as some of *septica*, subspecies identifications mentioned in the GenBank file headers. However, the *in-silico* PCR did not predict a clade of isolates that were reported as subsp. *septica* in the headers. Conversely, the *in-silico* PCR predicted that most isolates from the deep-branched clade of the ML phylogenic tree may belong to the *septica* subspecies, in accordance with the information included in 3/85 GenBank files for this clade. As the results of phenotypic tests for sorbitol and dulcitol fermentation of most *P. multocida* isolates in this clade are unknown, their subspecies classification could not be confirmed.

## Australian isolates have low rates of resistance to antimicrobials

The results of AST by microdilution revealed that most isolates from this study were susceptible to many antimicrobials commonly used in veterinary therapeutic treatments of bacterial infections (Table 5). Phenotypic resistances to beta-lactams were rarely observed, with only 1/58, 0/58 and 0/58 tested isolates showing resistance to ampicillin, amoxicillin/clavulanate, and cephalexin, respectively and 2/10 bovine isolates were found to be resistant to tetracyclines.

Sequence analysis of the complete set of *P. multocida* genomes with ABRIcate revealed high numbers of antimicrobial resistance genes (ARGs) in isolates from USA (mainly in cattle) and China (mainly in pigs), with a preponderance of resistances to aminoglycosides and tetracyclines in various country and hosts (Fig 8). Moreover, resistance markers to macrolides were mainly found in cattle isolates from USA, and in a few isolates from Spain and Australia.

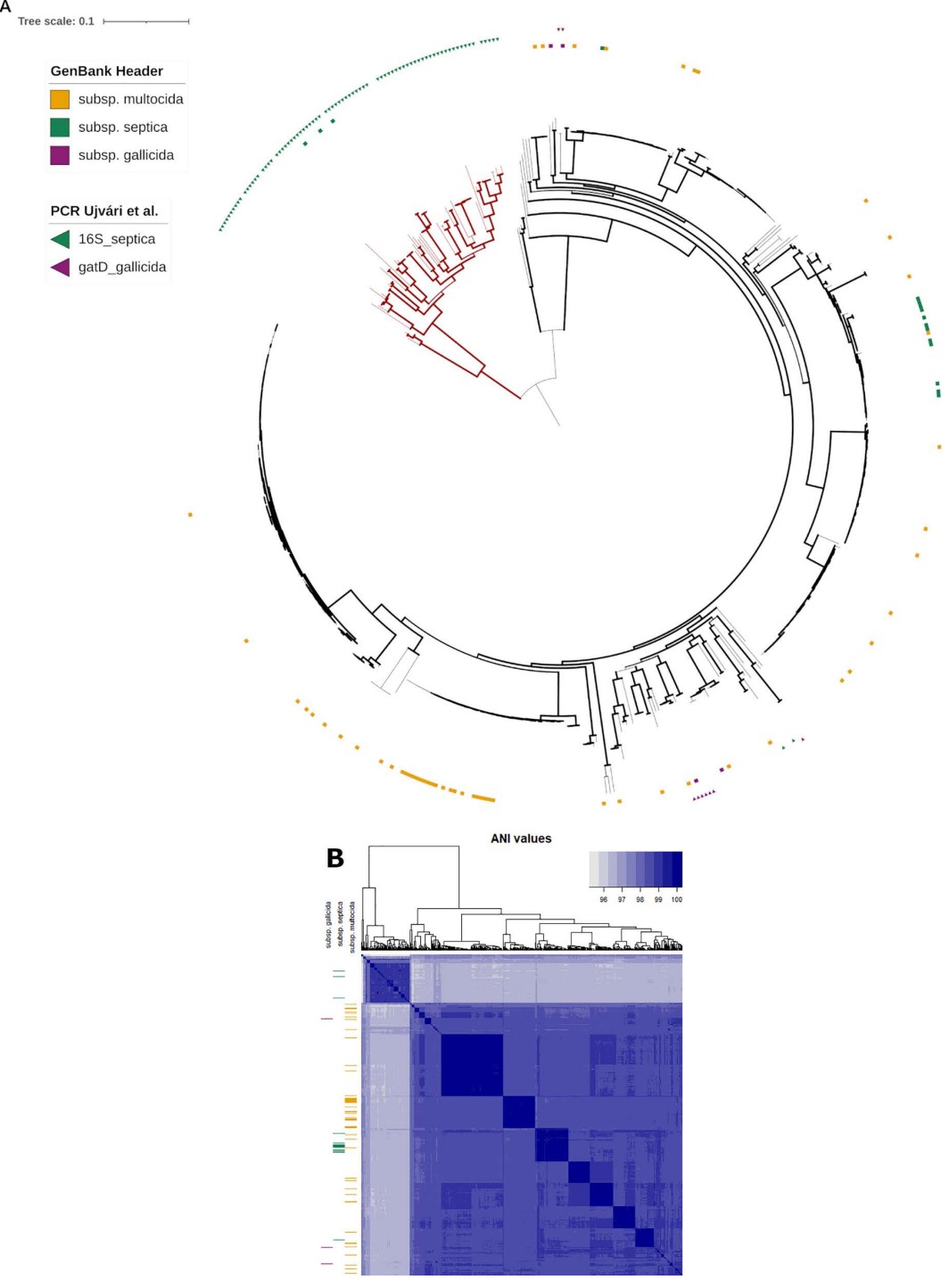

**Fig 7. Distribution of *P. multocida* subspecies within the population.** A set of 498 genomes from GenBank, supplemented by 84 Australian isolates sequenced in our laboratory (25 recently published ST20 and 59 from this study), was used. The colors yellow, green, and purple indicate the multocida, *septica,* or *gallicida* subspecies, respectively. **(A)** Unrooted maximum likelihood phylogenetic tree built from 585 core SNPs alignments of genomes, as shown in Fig 5. The deep-branched clade is indicated in red. Subspecies mentioned in the GenBank file header are reported by squares. Positive *in-silico* subspecies PCR results are indicated by triangles. **(B)** Hierarchical clustering of a matrix of pairwise ANI values calculated from the same set of 585 genomes. The left sidebar indicates the positions of the subspecies mentioned by the GenBank file headers.

**Table 5. Results of AST by microdilution for *P. multocida* isolates in this study.**

| Antimicrobial agent (concentration range[1]; (T) ECOFF[2]) | Observed MIC values (µg/mL) | Canine/Feline (n=19) | Chicken (n=17) | Ruminant (n=10) | Rabbit (n=5) | Captive wildlife (n=4) | Duck (n=1) | Porcine (n=1) | Environment (n=1) |
|---|---|---|---|---|---|---|---|---|---|
| Amoxycillin/clavulanic acid (0.25/0.12–8/4; 0.5) | 0.5/0.25 | 12 | 3 | 1 | 2 | – | 1 | – | 1 |
| | ≤0.25/0.12 | 7 | 14 | 9 | 3 | 4 | – | 1 | – |
| Ampicillin (0.25-8; 0.5) | 4 | – | – | – | – | – | 1 | – | – |
| | ≤0.25 | 19 | 17 | 10 | 5 | 4 | – | 1 | 1 |
| Piperacillin/tazobactam (8/4–64/4; ND) | ≤8/4 | 19 | 17 | 10 | 5 | 4 | 1 | 1 | 1 |
| Amikacin (4–32; ND) | 32 | 1 | – | – | – | – | – | – | – |
| | 16 | 4 | – | 1 | 2 | 2 | 1 | – | – |
| | 8 | 12 | 14 | 3 | 3 | – | – | – | 1 |
| | ≤4 | 2 | 3 | 6 | – | 2 | – | 1 | – |
| Gentamicin (0.5 - 8; 8) | 8 | – | – | – | – | 1 | – | – | – |
| | 4 | 3 | – | – | 1 | – | – | – | – |
| | 2 | 13 | 13 | 6 | 4 | 1 | 1 | 1 | 1 |
| | 1 | 2 | 4 | – | – | 2 | – | – | – |
| | 0.5 | 1 | – | 4 | – | – | – | – | – |
| Cefazolin (1–32; ID) | 2 | 1 | – | – | – | – | – | – | – |
| | ≤1 | 18 | 17 | 10 | 5 | 4 | 1 | 1 | 1 |
| Cefovecin (0.25–8; ND) | ≤0.25 | 19 | 17 | 10 | 5 | 4 | 1 | 1 | 1 |
| Cefopodoxime (1–8; ND) | ≤1 | 19 | 17 | 10 | 5 | 4 | 1 | 1 | 1 |
| Cephalexin (0.5–16; 8) | 4 | 7 | 2 | 1 | – | 1 | 1 | – | – |
| | 2 | 12 | 15 | 7 | 5 | 3 | – | 1 | 1 |
| | 1 | – | – | 1 | – | – | – | – | – |
| | ≤0.5 | – | – | 1 | – | – | – | – | – |
| Chloramphenicol (2–32; 1) | ≤2 | 19 | 17 | 10 | 5 | 4 | 1 | 1 | 1 |
| Imipenem (1–8; ID) | ≤1 | 19 | 17 | 10 | 5 | 4 | 1 | 1 | 1 |
| Enrofloxacin (0.12 - 4; 0.06) | ≤0.12 | 19 | 17 | 10 | 5 | 4 | 1 | 1 | 1 |
| Marbofloxacin (0.12–4; ND) | ≤0.12 | 19 | 17 | 10 | 5 | 4 | 1 | 1 | 1 |
| Orbifloxacin (1–8; ND) | ≤1 | 19 | 17 | 10 | 5 | 4 | 1 | 1 | 1 |
| Pradofloxacin (0.25–2; ND) | ≤0.25 | 19 | 17 | 10 | 5 | 4 | 1 | 1 | 1 |
| Tetracycline (4–16; 2) | 16 | – | – | 2 | – | – | – | – | – |
| | ≤4 | 19 | 17 | 8 | 5 | 4 | 1 | 1 | 1 |
| Doxycycline (0.25–8; 1) | 2 | – | – | 2 | – | – | – | – | – |
| | 1 | – | 2 | – | – | – | – | 1 | – |
| | 0.5 | 1 | | 1 | – | – | – | – | 1 |
| | ≤0.25 | 18 | 15 | 7 | 5 | 4 | 1 | – | – |
| Trimethoprim/ sulfamethoxazole (0.5/9.5–4/76; 0.125) | ≤0.5/9.5 | 19 | 17 | 10 | 5 | 4 | 1 | 1 | 1 |

[1]Concentration range (µg/mL) tested using the Sensititre COMPGN1F plate

[2]Consensus (T)ECOFF value (µg/mL) as reported by EUCAST, MIC distributions for *P. multocida* (2024-08-06) https://mic.eucast.org/search/; ND: no data; ID: insufficient data. Values observed above the ECOFF value are highlighted in red.

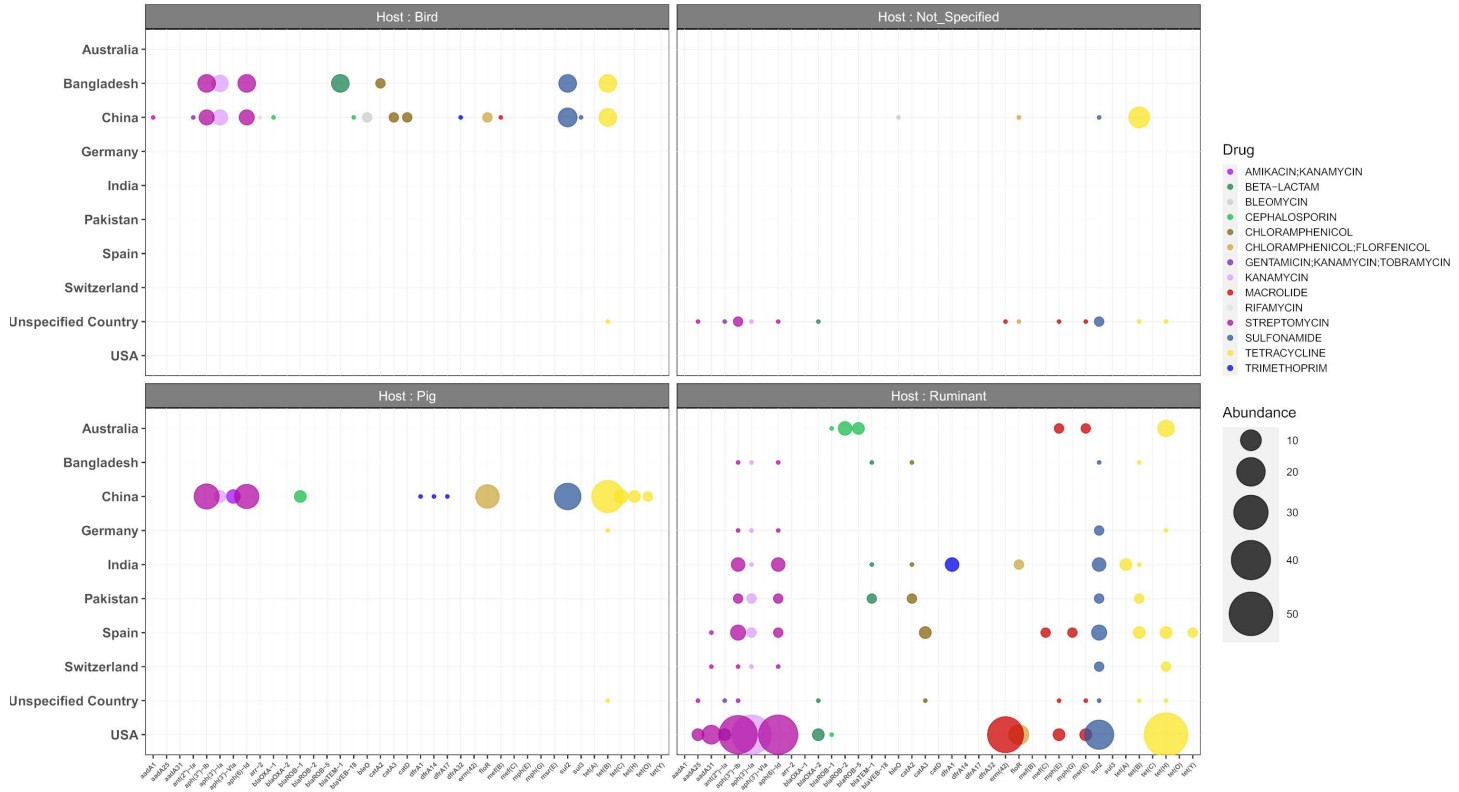

**Fig 8. Predicted ARGs in *P. multocida* genomes from compiled ABRIcate results.** Resistances to aminoglycosides, beta-lactams, phenicols, macrolides, tetracyclines and trimethoprim/sulfamides are respectively represented by purple, green, brown, red, yellow and blue color dots, with sizes proportional to the number of hits within the dataset.

By contrast, ARGs were detected only in ST394 isolates (clonal complex ST124) from bovine hosts located in Australia (Table 6). Four isolates from Queensland (19BRD-057 and −032, 18BRD-001 and 17BRD-035) carried extended spectrum beta-lactamase (ESBL) and tetracycline resistance genes on their chromosome, near a putative conjugative transposon. Two recent isolates from Victoria (CM2023-0122-0 and CN2023-222-2), also belonging to ST394, lacked the beta-lactamase genes but possessed ARGs conferring resistance to macrolides namely *msr*(E) and *mph*(E), as well as tetracyclines, namely *tet*(H). These ARGs were located separately on two circularized contigs of 7.6 and 4.6 kb, respectively, identified as plasmids from *Pseudomonadales* by the program PLASMe with scores over 0.99. The 7.6 kb plasmid was detected only in the bovine isolates from Victoria, and was related to a ~ 20 kb plasmid (Acc. NZ_CP048662.1) from *Acinetobacter piscicola* strain YH12207_T, which carried the same predicted markers of resistance to macrolides and replication initiation gene, but had a more complex organization (Fig 9).

The 4.6 kb plasmid sequence was related to a large conjugative plasmid, NZ_CP024444.1 from *Moraxella osloensis* strain NP7 carrying multiple ARGs, including *tet*(H). This sequence was also detected by PLASMe in two other *P. multocida* genomes from China (Acc. GCF_013013585.1) and Bangladesh (GCF_013013585.1). However, unlike the Australian genomes, the sequences were not circularized in the assembly files, and only the genome from China was predicted to contain a *tet*(H) copy. Furthermore, all genomes were systematically searched for a C1194G transversion in their 16S rRNA genes conferring resistance to spectinomycin. This mutation was detected in only 2 genomes from China (Ac. GCF_013013645.1 and GCF_013013535.1), and in none of the other isolates.

**Table 6. Predicted ARGs in Australian bovine *P. multocida* ST394 genomes.**

| Source | Year | Accession | Strain | ARG | ARG product | Conferred pheno-typic resistance | % Identity |
|---|---|---|---|---|---|---|---|
| Gen-Bank | 2019 | GCF_029873295.1 | 19BRD-057 | blaROB-5 | class A beta-lactamase ROB-5 | Cephalosporin | 99.78 |
| | | | | blaROB-2 | class A beta-lactamase ROB-2 | Cephalosporin | 99.89 |
| | | | | tet(H) | tetracycline efflux MFS trans-porter Tet(H) | Tetracycline | 100 |
| | 2019 | GCF_029906265.1 | 19BRD-032 | blaROB-5 | class A beta-lactamase ROB-5 | Cephalosporin | 99.78 |
| | | | | blaROB-2 | class A beta-lactamase ROB-2 | Cephalosporin | 99.89 |
| | | | | tet(H) | tetracycline efflux MFS trans-porter Tet(H) | Tetracycline | 100 |
| | 2018 | GCF_029906285.1 | 18BRD-001 | blaROB-5 | class A beta-lactamase ROB-5 | Cephalosporin | 99.78 |
| | | | | blaROB-2 | class A beta-lactamase ROB-2 | Cephalosporin | 99.89 |
| | | | | tet(H) | tetracycline efflux MFS trans-porter Tet(H) | Tetracycline | 100 |
| | 2017 | GCF_032027925.1 | 17BRD-035 | blaROB-1 | cephalosporin-hydrolyzing class A beta-lactamase ROB-1 | Cephalosporin | 99.78 |
| | | | | blaROB-2 | class A beta-lactamase ROB-2 | Cephalosporin | 100 |
| | | | | tet(H) | tetracycline efflux MFS trans-porter Tet(H) | Tetracycline | 100 |
| This study | 2023 | in progress | CM2023-0122-0 | mph(E) | Mph(E) family macrolide 2'-phosphotransferase | Macrolide | 99.89 |
| | | | | msr(E) | ABC-F type ribosomal protec-tion protein Msr(E) | Macrolide | 99.93 |
| | | | | tet(H) | tetracycline efflux MFS trans-porter Tet(H) | Tetracycline | 100 |
| | 2023 | in progress | CM2023-0222-2 | mph(E) | Mph(E) family macrolide 2'-phosphotransferase | Macrolide | 99.89 |
| | | | | msr(E) | ABC-F type ribosomal protec-tion protein Msr(E) | Macrolide | 99.93 |
| | | | | tet(H) | tetracycline efflux MFS trans-porter Tet(H) | Tetracycline | 100 |

Overall, very few ARGs were detected by sequence analysis of Australian *P. multocida* genomes, confirming the trend identified by phenotypic testing. The absence of resistance was particularly noted in isolates from pet dogs, cats and rabbits, which represented a large proportion of this study.

## Discussion

The genomic diversity of *P. multocida* has been rarely investigated in clinical isolates from companion animals. This is likely because public databases of partially or fully assembled sequences for this pathogen are skewed towards production animals, often with country-dependent distributions. For example, the swine isolates deposited in GenBank come preponderantly from China, (which is unsurprising considering that Asia is a major producer of pork worldwide), but are underrepresented in data from Europe and North America, despite the presence of a pig industry [31] and evidence of porcine infections [32,33] within these two regions of the world. In Australia, genomic datasets are mostly related to fowl cholera outbreaks in commercial poultry [34,35]. By contrast, publicly available complete genome sequences from non-avian hosts in Australia are rare, precluding phylogenomic and epidemiologic comparative investigations with other

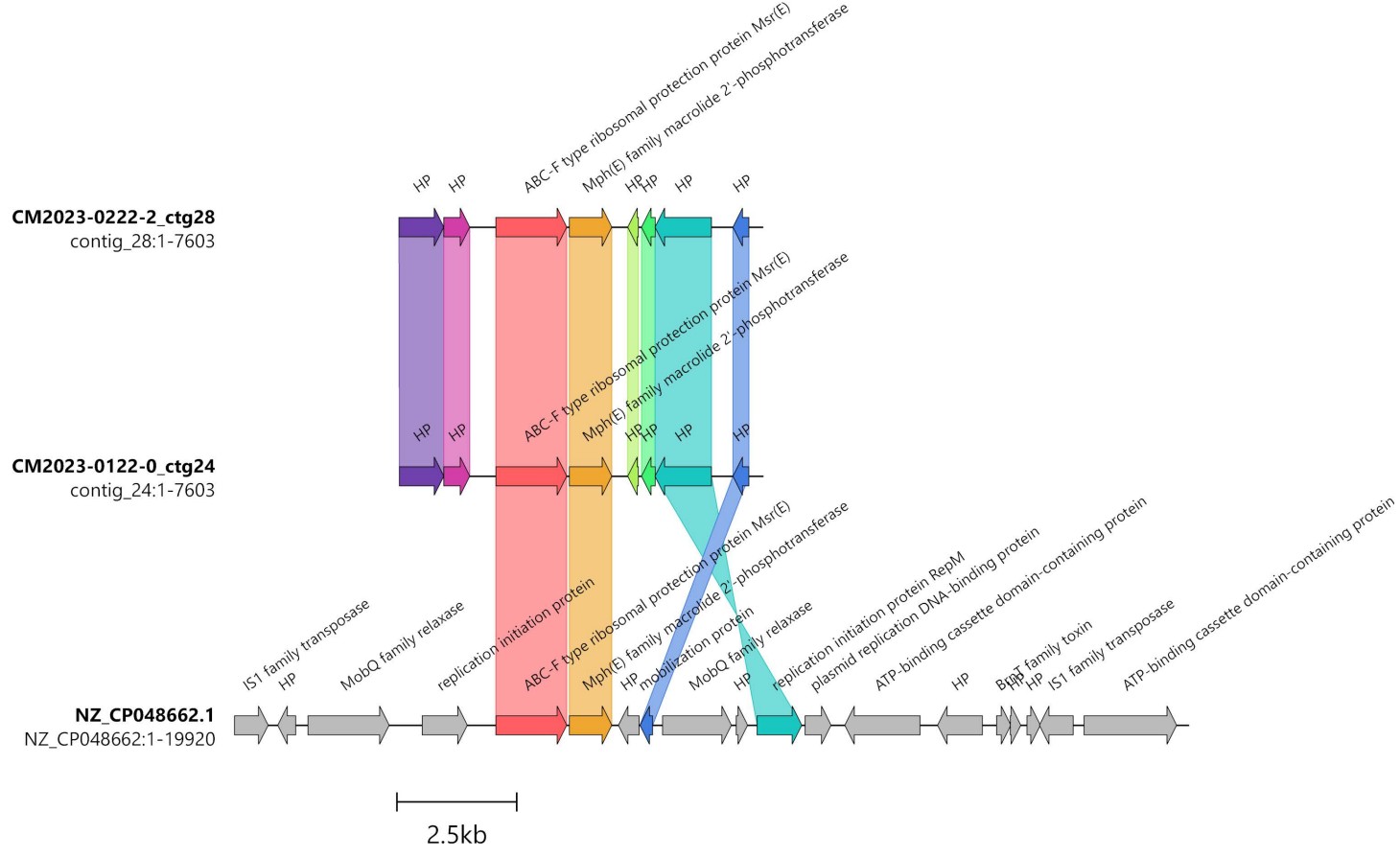

**Fig 9. Alignment of the 7.6 kb circular contigs found in two bovine isolates from Victoria and the plasmid pYH12207−3 of *Acinetobacter piscicola* strain YH12207_T. Genes belonging to the identical similarity group are represented by the same color.**

countries. Here, we have expanded the Australian repertoire of *P. multocida* hosts, with a particular emphasis on companion animals, by sequencing 25 new isolates from cats, dogs and pet rabbits, a substantial increase from the 17 isolates from cats and dogs, of which 14 came from Australia, found in GenBank prior to this study. Nevertheless, further efforts to publicly share genomes from a wider range of host species and geographical origins are still needed.

The intra-specific genomic diversity of *P. multocida* across various continents has been explored recently [7]. However, most isolates used in this study were from North America, with much smaller numbers from Asia, Europe and Australia. Moreover, the majority of the Australian data (75/95 genomes) related to a single free-range chicken farm located in the state of Queensland [35], while the remainder were from only two wildlife types (otariids and marsupials). The recent release of Australian human and animals genomes [12] has expanded the scope for comparative genomics of *P. multocida*, allowing for a deeper analysis of the dataset presented here. These two studies have indicated the presence of a unique long-branched clade within the *P. multocida* phylogenetic tree, different from the other clades by more than 6000 SNPs. Here, we have confirmed this observation, and provided new insight into the genomic diversity of this particularly intriguing clade. Indeed, 24/25 isolates from Australian pets described in our study were also located within this long-branched clade, of which 8 had a novel MLST profile recently added to the PubMLST database, namely ST451 and ST527. The predicted amplification of a PCR product from primers designed to identify the subspecies *septica* [21] from most genomic sequences of this clade warrants a careful evaluation of the taxonomic rank in this phylogenic group,

which is beyond the aims of the present report. Furthermore, the same clade contained several poultry farm isolates with STs rarely or never reported in Australia (i.e., ST183, described so far only in Germany in 1988 and USA in 2019); by contrast, other avian isolates with MLST groups commonly reported in Australian broilers or layers, such as ST8, ST9, or ST20 [13,34,35], fell into the main section of the tree. Thus, this long-branched clade appears to represent a distinct subset of the *P. multocida* population, showing specific epidemiologic properties and potential evidence of cross-species or zoonotic transmissions. This is exemplified by two nearly identical strains collected in 2016 in Greece (NZ_CM009574 and NZ_PSQI01000001), isolated from a dog and a human patient, respectively. Such a finding is consistent with the fact that *P. multocida* is one of the most common contributors of zoonotic infections following bites or scratches by pets [36]. In Australia, while some studies have examined human infections caused by *P. multocida* [37,38], until recently there has been little or no associated genomic information on these isolates, and it remains difficult to trace suspect zoonotic transmissions. The recent release in GenBank of 22 Australian human isolates [12] has allowed us to identify highly related genomes in our collection. The low number of SNPs (i.e., 46–52) and the small goeBURST distances, suggest a quasi-clonal relation between these isolates, which came from a liver tissue of a farmed chicken in 2009, a human skin wound in 2017, and a thoracic tissue mass of a dog in 2020. Other suspect transmissions between humans and pets in Australia were tentatively identified for two cat isolates, one from an abdominal fluid in 2019 and one from a pleural infection in 2021, both having their closest relative in a human associated genome, respectively from 2014 and 2017, and both displaying approximately 220–230 SNPs compared to their human counterpart. While such numbers of SNPs are likely to be too high to indicate a clonal relationship, it must be noted that the output from snippy might be influenced by possible recombination, therefore may underestimate the true proximity of these isolates. In addition, the identification of pairs of highly related Australian genomes which share as few as 30–50 SNPs within each set, despite being isolated from unrelated animal sources (cats and rabbits) strongly suggests that cross-species transmission of *P. multocida* is not uncommon.

It is unclear how these strains infected their hosts and were transferred to a new host, as no commonality in the case histories is evident, apart from the fact that all isolates originate from Australia. Therefore, our study illustrates the need for wider investigations of cross-species, and potentially zoonotic, pasteurellosis in Australian pets, production animals, and humans. This is particularly relevant as a wide array of MGEs were found to be present in Australian *P. multocida* genomes, and resistance plasmids found in other taxonomic groups were detected in recent bovine isolates. An intriguing observation from this work was the detection of a cryptic plasmid U51470 in a significant number of isolates which were quasi-exclusively from Australia, but close to none of over 400 genomes from the rest of the world. Since this plasmid is also found in *P. canis*, a rare but possible human pathogen [39], it would be interesting to investigate the distribution and significance of this replicon in other members of the *Pasteurella* genus.

Beyond public health concerns, our study highlights the importance of biosecurity in modern farming systems. Close genomic relatedness has been recently described between ST20 isolates from wild waterbirds and several free-range poultry farms [13], opening the question of spill-over between free-ranging animal populations. Here, the presence in the long-branched clade of two highly similar ST23 genomes, in a duck and a chicken from different farms, demonstrates that *P. multocida* from avian hosts belonging to different taxonomic orders, i.e., *Anseriformes* (which include waterfowls like ducks) and *Galliformes* (which include land fowls such as chickens and turkeys) can share common ancestors. Unfortunately, the presence of *P. multocida* in potential wildlife reservoirs such as free-ranging rodents, foxes, rabbits and feral cats, is not documented in the current datasets of complete genomes from purified isolate, although metagenomic studies provide indirect evidence of the colonization of oral cavity of cats by the microorganism [40,41]. This lack of information may constitute a critical epidemiological blind-spot on pasteurellosis, particularly in view of the well-known roles of foxes and feral cats as predators for wildlife in Australia [42].

Antimicrobial resistance profiles in *P. multocida* from pets can vary significantly with the geographic origin of the isolates [43–45]; remarkably, the phenotypic and genotypic characterization of Australian isolates from companion animals

showed a wide susceptibility to common veterinary antimicrobials. Moreover, our study found only a small number of ARGs in ruminant isolates, mainly related to low importance antimicrobials, and none in avian strains. A similar trend of low prevalence of ARGs has been suggested for Australian pig isolates [46]. These local susceptibility results offer some contrast with other parts of the world [47,48], and may be explained by the strict control of antimicrobial use in production animals in Australia. However, macrolide and tetracycline resistant *P. multocida* emerging in Australian BRD affected feedlot cattle may indicate management practices within production animal systems also contribute to observed patterns of antimicrobial resistance [49].

In conclusion, the present study has considerably expanded the diversity of origins of *P. multocida* genome sequences currently deposited in public databases. This new dataset is expected to provide a solid base for future comparative phylogenomic investigations on this important veterinary and zoonotic pathogen.

## Supporting information

**S1 Table. Details of *P. multocida* isolates from this study.**
(PDF)

**S2 Table. Details of the published *P. multocida* genomes used in comparative analyses within this study.**
(PDF)

**S3 Table. Genome sequencing statistics for *P. multocida* isolates from this study.**
(PDF)

**S4 Table. List of the 94 RefSeq genomes from the Pasteurellaceae family used to confirm taxonomic identification.**
(PDF)

**S5 Table. Country and host distribution of whole genome sequences of *P. multocida* analysed with PopPUNK.**
(PDF)

**S6 Table. GoeBURST groups with less than 4 loci differences.** Refer to Table 2 for the five subsets (A-E) containing mixed host species.
(PDF)

## Author contributions

**Conceptualization:** Joanne L. Allen, Amir H. Noormohammadi, Pam Whiteley, Glenn F. Browning, Marc S. Marenda.

**Data curation:** Joanne L. Allen, Rhys N. Bushell, Susan A. Ballard, Mary Valcanis, Marc S. Marenda.

**Formal analysis:** Joanne L. Allen, Susan A. Ballard, Mary Valcanis, Marc S. Marenda.

**Investigation:** Joanne L. Allen, Rhys N. Bushell, Marc S. Marenda.

**Methodology:** Joanne L. Allen, Rhys N. Bushell, Susan A. Ballard, Mary Valcanis, Marc S. Marenda.

**Project administration:** Joanne L. Allen, Susan A. Ballard, Marc S. Marenda.

**Resources:** Amir H. Noormohammadi, Susan A. Ballard, Mary Valcanis, Glenn F. Browning, Marc S. Marenda.

**Software:** Susan A. Ballard, Mary Valcanis, Marc S. Marenda.

**Supervision:** Amir H. Noormohammadi, Glenn F. Browning, Marc S. Marenda.

**Validation:** Joanne L. Allen, Rhys N. Bushell, Susan A. Ballard, Mary Valcanis, Marc S. Marenda.

**Visualization:** Joanne L. Allen, Rhys N. Bushell, Mary Valcanis, Marc S. Marenda.

**Writing – original draft:** Marc S. Marenda.

**Writing – review & editing:** Joanne L. Allen, Rhys N. Bushell, Amir H. Noormohammadi, Pam Whiteley, Susan A. Ballard, Mary Valcanis, Glenn F. Browning, Marc S. Marenda.

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
