## [Decision Letter · Decision Letter 0]

15 Sep 2024

PONE-D-24-33400Comparative genome analysis of *Pasteurella multocida* from Australian domestic animals suggests broad patterns of transmissions across multiple hosts and originsPLOS ONE

Dear Dr. Allen,

Thank you for submitting your manuscript to PLOS ONE. After careful consideration, we feel that it has merit but does not fully meet PLOS ONE’s publication criteria as it currently stands. Therefore, we invite you to submit a revised version of the manuscript that addresses the points raised during the review process.

We look forward to receiving your revised manuscript.

Kind regards,

Mahmoud Abdel Aziz Mabrok, PhD

Academic Editor

PLOS ONE

Additional Editor Comments:

This article is well-suited for publication, and the authors have made a commendable effort in writing and presenting the conclusions. However, some revisions are necessary before it can be published. Please address the reviewer comments and queries, and make the necessary corrections and modifications

Reviewers' comments:

Reviewer's Responses to Questions

**Comments to the Author**

1. Is the manuscript technically sound, and do the data support the conclusions?

Reviewer #1: Partly

Reviewer #2: Yes

Reviewer #3: Yes

2. Has the statistical analysis been performed appropriately and rigorously? 

Reviewer #1: N/A

Reviewer #2: N/A

Reviewer #3: Yes

3. Have the authors made all data underlying the findings in their manuscript fully available?

Reviewer #1: No

Reviewer #2: No

Reviewer #3: Yes

4. Is the manuscript presented in an intelligible fashion and written in standard English?

Reviewer #1: Yes

Reviewer #2: Yes

Reviewer #3: Yes

5. Review Comments to the Author

Reviewer #1: The reviewed manuscript aimed to comparative analysis of whole genomes of Pasteurella multocida strains isolated from companion animals in Australia with those available in public databases. The authors presented very interesting and important study with the strains which were recently characterized by this research group (ref 12 in the present manuscript).

Specific comments:

1. Line 26 - Pasteurella multocida (Pm) could infect not only animals as the authors cites in the section Introduction (lines 51-53), please, correct;

2. Lines 29-30 – healthy pets and other animals? if not, clinical diagnosis should be indicated. Alternatively, the reason for the current bacterial investigation must be included. Additionally for captive wildlife - are their zoo animals?

3. Line 36 - novel STs should be listed;

4. Lines 40-42 – how much? n=….(% from totally investigated Australian P.

multocida isolates);

5. Line 43 – similarly, the number of isolates with AMR (n=…, %) and the AMR spectrum detected in isolates from Australian companion animals must be added;

6. Line 49 – please, include ‘companion animal’, ‘Pasteurella multocida’, ‘ST’, ‘whole genome’ etc. to the keywords to make your research more available for search by worldwide scientific community;

7. Lines 86-93 – more information on the Pm isolates must be added, i.e., the strains were isolated from healthy animals? Any pasteurellosis cases in animals/humans/owners were related/ associated? Did any other either bacterial or viral strains were isolated from the same specimens?

8. Line 144 - the accession numbers need to be check to make them available for any potential reader;

9. Which Pm subspp. was finally identified? This information should be clearer indicated through the manuscript including the section Abstract;

10. The legend for Figure 2 needs to include more details in description; alternatively, Figures 2 and 3 could be companied in a single Figure;

11. Lines 235-238 – the concatenated sequences for housekeeping genes should be deposited in PubMLST database to determine the relevant ST for each of the Australian isolates with unclassified STs, the data should be added to the Table 1; the alleles with mutation could be highlighted to make these data more visible for potential readers;

12. The data presented in the section ‘Phylogeny and MLST strongly suggest multiple instances of cross-species transmission in companion animals’ needs to more evidences to make the conclusions concerning a cross-species transmission of Pm from companion animals to other host spp. No information was presented on clinical cases in owners & zoo/farm personnel & closely living or contacting wild/domestic animals etc. and companions with Pm-related infection following identification of the identical Pm isolates & genomes. Similar Pm strains are isolated from different hosts being adapted for different hosts. However, no information on either direct or indirect cross-species transmission, i.e. transmission the Pm strain from one host to another;

13. Line 298 – please, indicate in footnote the source for the information presented in Table 3;

14. Lines 335 through the section AMR – no complete data on AMR for Pm strains investigated could be received, if some genomes were analyzed as contigs;

15. the results demonstrated the presence of AMR in the Pm strains should be highlighted to make the data presented in Table 4 more clear for potential readers;

16. Lines 457-458 – P. multocida is very important pathogen for both veterinary and Public Health worldwide. More than 1,000 papers published could be found easily: https://pubmed.ncbi.nlm.nih.gov/?term=Pasteurella+multocida+AND+human+infection&sort=date.

Reviewer #2: The manuscript presents a genomic and phylogenetic analysis of Pasteurella multocida isolates, with a focus on Australian companion animals. It is a well-written study that significantly contributes to understanding the genomic diversity of this pathogen. However, it overlooks several key publications, particularly recent genomic studies from Australia. While I understand that some of these publications on P. multocida from human and companion animals were released after the authors' NCBI GenBank access date (December 13, 2023), it is unfortunate that they will need to redo portions of their analysis. As their work focuses on Australia, incorporating this new data would strengthen the manuscript's relevance and provide a more comprehensive discussion of the pathogen in this region. Below, I provide a detailed review with recommendations to include the new genomic data, improve clarity, and refine specific analyses.

The manuscript claims there were no genome from Australian companion animals at the date of NCBI GenBank database access. However, genomes of several P. multocida isolates from human and companion animals from Australia have been analysed and deposited in public database. Given the manuscript’s focus on Australian isolates, I recommend incorporating this recent data to update the results and discussion.

Specifically, the following publication available from March 1, 2024 could significantly inform the current analysis:

Smallman TR, Perlaza-Jiménez L, Wang X, Korman TM, Kotsanas D, Gibson JS, et al. Pathogenomic analysis and characterization of Pasteurella multocida strains recovered from human infections. Microbiol Spectr. 2024;12(4):e0380523. (available from 1st of March 2024)

Furthermore, the authors may want to consider incorporating data from a recently published complete genome of a P. multocida isolate from an Australian pig

Chau D, Blackall PJ, Turni C, Omaleki L. Complete genome sequence of a Pasteurella multocida isolate from a pig in Australia. Microbiology resource announcements.0(0):e00241-24.

Result section:

Figures 2 and 3: The phylogenetic trees are similar to those described in the Smallman et al. publication, with a distinct deep-branched cluster. However, the quality of the images in this manuscript made it difficult to access the phylogenetic relationships. To improve the clarity and robustness of the phylogenetic analysis, I suggest comparing these results with those from Smallman et al., which may offer additional insights and reinforce the authors' conclusions.

Line 255: The manuscript suggests that close phylogenetic relatedness and identical MLST profiles indicate host transfers. While this is possible, it is not definitive. Tools such as PHYLOViZ could be used to more accurately represent evolutionary relationships between strains. For example, the chicken isolate (CM2007-0542-0) and duck isolate (CM2013-0823-1) share a common ancestor but this does not necessarily imply direct transfer from one host to the other.

Line 260: The manuscripts states that Australian isolates PM1541 and PM1582 belong to ST451. However, according to the original publication, these isolates were identified as ST274 (both based on RIRDC scheme). I suggest the authors revisit the data to confirm the correct sequence type.

Line 262: The Smallman TR et al., publication also demonstrates the clustering of the USA strain FDAARGOS_384 and an Australian human isolate. Comparing the findings from these two manuscripts would provide further insight into global strain distribution and zoonotic transmission patterns.

Line 317: The observation that subspecies classification in published genomes does not match phylogeny closely is an important point. To explore this further, I recommend an in-silico approach using available genomic data to differentiate between subspecies. Ujvári et al. (2022) provide a framework for identifying key genes associated with Pasteurella multocida subspecies, which could be useful for this reanalysis. This method would enable verification of the GenBank metadata and address potential misclassifications without the need for additional laboratory work.

Ujvári, B., Gantelet, H., Magyar, T., 2022. Development of a multiplex PCR assay for the detection of key genes associated with Pasteurella multocida subspecies. J. Vet. Diagn. Invest. 34, 319-322.

I also suggest incorporating findings from Alhamami et al. (2023), which focus on antimicrobial resistance genes (ARG) in P. multocida isolates from Australian feedlot cattle. This would enhance the discussion around ARG in P. multocida originated from various host species.

Alhamami T, Roy Chowdhury P, Venter H, Veltman T, Truswell A, Abraham S, et al. Genomic profiling of Pasteurella multocida isolated from feedlot cases of bovine respiratory disease. Vet Microbiol. 2023;283:109773.

The mentioned Illumina reads can not located in PRJNA965909

Reviewer #3: This article is very good for publication, and the authors have put forth their best effort in writing and presenting the conclusion. However, some changes are required before its publication. I have mentioned all the points below:

I suggest authors to read and incorporate the information from the following articles and cite them:

mRNA vaccines as an armor to combat the infectious diseases. Travel Medicine and Infectious Disease 52:102550.

Zoonotic diseases in a changing climate scenario: Revisiting the interplay between environmental variables and infectious disease dynamics, Travel Medicine and Infectious Disease,

58:102694.

Nanovaccines: A game changing approach in the fight against infectious diseases. Biomedicine & Pharmacotherapy 167(2023):115597

6. PLOS authors have the option to publish the peer review history of their article (what does this mean? ). If published, this will include your full peer review and any attached files.

**Do you want your identity to be public for this peer review?** For information about this choice, including consent withdrawal, please see our Privacy Policy .

Reviewer #1: No

Reviewer #2: No

Reviewer #3: No

---

## [Author Response · Author response to Decision Letter 1]

17 Nov 2024

We had attached a rebuttal letter that systematically responds to each of the comments made by the Academic Editor and the 3 reviewers.

---

## [Decision Letter · Decision Letter 1]

14 Jan 2025

PONE-D-24-33400R1Comparative genome analysis of *Pasteurella multocida* from Australian domestic animals suggests broad patterns of transmissions across multiple hosts and originsPLOS ONE

Dear Dr. Allen,

Thank you for submitting your manuscript to PLOS ONE. After careful consideration, we feel that it has merit but does not fully meet PLOS ONE’s publication criteria as it currently stands. Therefore, we invite you to submit a revised version of the manuscript that addresses the points raised during the review process.

We look forward to receiving your revised manuscript.

Kind regards,

Mahmoud Abdel Aziz Mabrok, PhD

Academic Editor

PLOS ONE

Journal Requirements:

Reviewers' comments:

Reviewer's Responses to Questions

**Comments to the Author**

1. If the authors have adequately addressed your comments raised in a previous round of review and you feel that this manuscript is now acceptable for publication, you may indicate that here to bypass the “Comments to the Author” section, enter your conflict of interest statement in the “Confidential to Editor” section, and submit your "Accept" recommendation.

Reviewer #1: (No Response)

Reviewer #2: All comments have been addressed

2. Is the manuscript technically sound, and do the data support the conclusions?

Reviewer #1: Partly

Reviewer #2: Yes

3. Has the statistical analysis been performed appropriately and rigorously? 

Reviewer #1: N/A

Reviewer #2: N/A

4. Have the authors made all data underlying the findings in their manuscript fully available?

Reviewer #1: No

Reviewer #2: Yes

5. Is the manuscript presented in an intelligible fashion and written in standard English?

Reviewer #1: Yes

Reviewer #2: Yes

6. Review Comments to the Author

Reviewer #1: The authors of the reviewed manuscript made the majority necessary corrections while they must be sure that all the data presented here are deposited in PubMLST, NCBI and other public repository. First of all, ST-related nucleotide sequences should be mandatory deposited in PubMSLT database and available for potential readers of the Journal according to the general policy of the Plos One (see the answer to my relevant recommendations: 'Our intention is to submit the full genomic sequences to the PubMLST database after publication. It takes some time for a new sequence type to be assigned, and we have decided to precede with publication rather than wait.'). Currently, all data underlying the findings in their manuscript could not considered as ‘fully available’. No additional comments.

Reviewer #2: Thanks to the authors for taking my suggestion to consideration. Please see below minor comments to be addressed before publication.

Fig 1. I can not find ST 20 on the x axis. Based on the manuscript, ST 20 consists of a unique population only found in Australia.

Fig 4. There seems to be a disagreement between the Fig legend in regards to the color strip. From left, they are host category, capsule type and LPS type.

Fig 5, Table 2 and the relevant text can benefit from a little more explanation for making it clear to the reader. First, the authors have not mentioned how many groups they recognised in their phyloviz analysis. Assuming the goeBURST A-E in Table 2 is referring to that grouping, groups C, D and E are not marked on Fig 5. Maybe adding a footnote to Table 2 mentioning that the Chestnut_Teal and Swan isolates are depicted by yellow circle in the Phyloviz analysis makes it easier for the reader to follow. For other groups, I can not find any way for the reader to locate those isolates on Fig 5 and one should simply deal with that.

7. PLOS authors have the option to publish the peer review history of their article (what does this mean? ). If published, this will include your full peer review and any attached files.

**Do you want your identity to be public for this peer review?** For information about this choice, including consent withdrawal, please see our Privacy Policy .

Reviewer #1: No

Reviewer #2: No

---

## [Author Response · Author response to Decision Letter 2]

26 Jan 2025

We would like to thank the editor and reviewers for the contributions that they have made to improve the quality of our paper. We have provided a separate file where we have detailed how we have addressed their comments about our previous submission.

---

## [Decision Letter · Decision Letter 2]

10 Apr 2025

PONE-D-24-33400R2Comparative genome analysis of *Pasteurella multocida* from Australian domestic animals suggests broad patterns of transmissions across multiple hosts and originsPLOS ONE

Dear Dr. Allen,

Thank you for submitting your manuscript to PLOS ONE. After careful consideration, we feel that it has merit but does not fully meet PLOS ONE’s publication criteria as it currently stands. Therefore, we invite you to submit a revised version of the manuscript that addresses the points raised during the review process.

**ACADEMIC EDITOR:**

The authors ignore Reviewer 1's request to make fully available the sequences described in their manuscript. First of all, ST-related nucleotide sequences should be mandatorily deposited in the PubMSLT database and available for potential readers of the Journal according to the general policy of Plos One. Several revisions are required, including structure enhancements, clarity, and scientific rigor.

We look forward to receiving your revised manuscript.

Kind regards,

Faham Khamesipour, Ph.D.

Academic Editor

PLOS ONE

Journal Requirements:

Additional Editor Comments:

The authors ignore Reviewer 1's request to make fully available the sequences described in their manuscript. First of all, ST-related nucleotide sequences should be mandatorily deposited in the PubMSLT database and available for potential readers of the Journal according to the general policy of PLOS One. Several revisions are required, including structure enhancements, clarity, and scientific rigor.

Reviewers' comments:

Reviewer's Responses to Questions

**Comments to the Author**

1. If the authors have adequately addressed your comments raised in a previous round of review and you feel that this manuscript is now acceptable for publication, you may indicate that here to bypass the “Comments to the Author” section, enter your conflict of interest statement in the “Confidential to Editor” section, and submit your "Accept" recommendation.

Reviewer #1: All comments have been addressed

Reviewer #4: (No Response)

2. Is the manuscript technically sound, and do the data support the conclusions?

Reviewer #1: Partly

Reviewer #4: Yes

3. Has the statistical analysis been performed appropriately and rigorously? 

Reviewer #1: Yes

Reviewer #4: N/A

4. Have the authors made all data underlying the findings in their manuscript fully available?

Reviewer #1: No

Reviewer #4: Yes

5. Is the manuscript presented in an intelligible fashion and written in standard English?

Reviewer #1: Yes

Reviewer #4: Yes

6. Review Comments to the Author

Reviewer #1: No additional comments. The authors should read my previous comments. In the current state, current form, not all data is fully accessible to potential readers.

Reviewer #4: This paper describes genomic and phylogenetic analyses of Pasteurella multocida strains, focussing on those isolated from Australian companion animals. It supports and extends recent work showing that isolates from domestic animals and human infections are genetically divergent from the better studied P. multocida strains that infect production animals such as chickens, cattle and pigs. This paper highlights likely cross species transmission events by identifying very closely related isolates from different species. This is an important identification, but in my opinion, can be strengthened by clearly identifying the numbers of SNPs that separate isolates in Table 2, better linking Fig. 5 and Table 2, and adding species designations to Table 3 (as outlined in my major comments below). Overall, this paper is well-written and adds significantly to the current understanding of P. multocida phylogeny and host specificity.

Note that I did not review this previously, so the authors have not previously had a chance to address my comments.

Major comments

1. For the new genome sequences (sequenced in this study) are the assembly statistics available somewhere?

2. Apologies if I am missing something, but I find it difficult to link the Fig. 5 Phyloviz goeBurst analysis and Table 2. Table 2 has GoeBurst groups of A, B, C, D and E but I can’t easily identify where these clusters are in Fig.5. Are these able to be highlighted somehow?

3. Table 2 indicates what are meant to be clonal complexes containing mixed host species. However, none of these appear to be in the Table 3, which shows closest SNP differences. Perhaps SNP differences (compared to the first strain in each group) could be added into Table 2. Was snippy used to compare all NGS reads from one strain against the reference genome of the other strain? Also, were default settings used to call SNPs?

4. Table 3 shows closely related strains (by SNP differences) that are likely associated with cross-species transmission. Can host species be added into this table to quickly identify those strains from different species.

5. I am unclear why you are unwilling to call the deeply separated clade a subspecies septica. The ANI matrix appears to show that clade, which I believe includes the subsp. septica type strain, as all having >98% identity and having less than 98% identity with the other larger multocida/gallicida clade. I think this is very strong evidence for these being septica. I agree that there is no evidence for separation of multocida/gallicida subspecies.

6. For Fig. 7b (the ANI matrix), the legend on the figure has 5 colors but 6 different numerical % values given, so they don’t match up in a standard way. This should probably just have <95.5, 95.5-96.5, 96.5-97.5, 97.5-98.5, 98.5-99.5, >99.5

Minor comments

1. Line 106; “read length 250x250 bp” is this meant to be 2x250?

2. Fig. 2. Even in the high-res version, the small text at 10 o’clock on the outer circles is nearly unreadable. I think it is just repeated in the legend text at bottom left. However, I think it would be better deleted (leaving just the large legend text) or increased in size to make it readable.

3. In Fig. 4 P1581 appears twice, once listed as avian and once listed as Pine Siskin.

7. PLOS authors have the option to publish the peer review history of their article (what does this mean? ). If published, this will include your full peer review and any attached files.

**Do you want your identity to be public for this peer review?** For information about this choice, including consent withdrawal, please see our Privacy Policy .

Reviewer #1: No

Reviewer #4: **Yes: ** John Dallas Boyce

---

## [Author Response · Author response to Decision Letter 3]

4 May 2025

We would like to thank the reviewers' for their valuable feedback and contributions. All reviewer comments for the third review have been addressed in the rebuttal letter, uploaded with this submission.

---

## [Decision Letter · Decision Letter 3]

19 Jun 2025

PONE-D-24-33400R3Comparative genome analysis of *Pasteurella multocida* from Australian domestic animals suggests broad patterns of transmissions across multiple hosts and originsPLOS ONE

Dear Dr. Allen,

Thank you for submitting your manuscript to PLOS ONE. After careful consideration, we feel that it has merit but does not fully meet PLOS ONE’s publication criteria as it currently stands. Therefore, we invite you to submit a revised version of the manuscript that addresses the points raised during the review process.

**ACADEMIC EDITOR:**

The reviewers recommend some revisions to ensure the robustness and validity of the findings before publication can be considered. I invite you to resubmit your manuscript and respond to each reviewer's comment by either outlining how the criticism was addressed in the revised manuscript or by providing a rebuttal.

We look forward to receiving your revised manuscript.

Kind regards,

Faham Khamesipour, Ph.D.

Academic Editor

PLOS ONE

Journal Requirements:

Reviewers' comments:

Reviewer's Responses to Questions

**Comments to the Author**

1. If the authors have adequately addressed your comments raised in a previous round of review and you feel that this manuscript is now acceptable for publication, you may indicate that here to bypass the “Comments to the Author” section, enter your conflict of interest statement in the “Confidential to Editor” section, and submit your "Accept" recommendation.

Reviewer #5: (No Response)

Reviewer #6: (No Response)

2. Is the manuscript technically sound, and do the data support the conclusions?

Reviewer #5: Yes

Reviewer #6: Yes

3. Has the statistical analysis been performed appropriately and rigorously? 

Reviewer #5: N/A

Reviewer #6: Yes

4. Have the authors made all data underlying the findings in their manuscript fully available?

Reviewer #5: Yes

Reviewer #6: Yes

5. Is the manuscript presented in an intelligible fashion and written in standard English?

Reviewer #5: Yes

Reviewer #6: Yes

6. Review Comments to the Author

Reviewer #5: After reviewing the original manuscript and the revised versions after earlier reviews, the manuscript has been significantly improved in the areas that were questioned. I have no objections to the revised manuscript.

Reviewer #6: This is a well-executed and innovative study that provides important insights into the genomic diversity of Pasteurella multocida in Australian domestic animals, including pets, farm animals, and captive wildlife. The work is highly relevant to the fields of microbiology and molecular epidemiology and stands out by highlighting potential cross-species and zoonotic transmission pathways, which are currently underexplored in the literature.

The comparative genomic approach, the inclusion of underrepresented host species, and the detailed phylogenomic and MLST analyses make this manuscript a valuable contribution to the understanding of P. multocida population structure and dynamics. The detection of clades associated with pets and humans, and the identification of shared sequence types across different hosts, open new perspectives for investigating zoonotic risks.

The manuscript is generally clear, well-written, and well-organized. The figures are informative and support the main findings appropriately.

Minor Suggestions:

1. If available, add more context regarding clinical outcomes or disease manifestations associated with isolates from companion animals and wildlife. The origin of the samples (e.g. respiratory or oral secretion, abscess, urine, wound secretions) should be mentioned, thus helping in the clinical and diagnostic conduct of veterinarians who work in the clinical care of these species. It can be included as a supplementary file (table).

2. It would be valuable to include a brief section outlining the limitations of the study. For example, challenges in sequencing quality, gaps in metadata (e.g., clinical presentation, host history), limited sample sizes for certain host species, or lack of functional validation for genomic predictions could be acknowledged. This would help contextualize the findings and guide future research directions.

3. Please consider including information on animal ethics approval. Even when using clinical or diagnostic samples from animals, ethical oversight by an institutional Animal Ethics Committee is generally required or at least advisable, and mentioning the approval protocol or exemption rationale is important for transparency.

4. If ethical approval was not obtained, we suggest that this be acknowledged as a limitation of the study. The lack of an ethics statement might raise questions regarding compliance with international standards for the ethical use of animals in research.

5. The discussion attributes the low antimicrobial resistance levels found in the isolates to the controlled and rational use of antimicrobials in Australian livestock, which is indeed commendable. However, an important question that arises is whether these findings could inform antimicrobial therapy choices for P. multocida infections in companion or production animals in other countries or regions. Addressing this point could enhance the broader applicability and relevance of the study and further support its alignment with the One Health approach. Strengthening this perspective may increase the impact of the work beyond the Australian context.

Overall, I strongly support the publication of this article after minor clarifications.

7. PLOS authors have the option to publish the peer review history of their article (what does this mean? ). If published, this will include your full peer review and any attached files.

**Do you want your identity to be public for this peer review?** For information about this choice, including consent withdrawal, please see our Privacy Policy .

Reviewer #5: No

Reviewer #6: No

---

## [Author Response · Author response to Decision Letter 4]

9 Jul 2025

We have provided a response to all reviewer comments in our rebuttal letter.

---

## [Decision Letter · Decision Letter 4]

22 Jul 2025

Comparative genome analysis of *Pasteurella multocida* from Australian domestic animals suggests broad patterns of transmissions across multiple hosts and origins

PONE-D-24-33400R4

Dear Dr. Allen,

We’re pleased to inform you that your manuscript has been judged scientifically suitable for publication and will be formally accepted for publication once it meets all outstanding technical requirements.

Kind regards,

Faham Khamesipour, Ph.D.

Academic Editor

PLOS ONE

Additional Editor Comments (optional):

Reviewers' comments:

Reviewer's Responses to Questions

**Comments to the Author**

1. If the authors have adequately addressed your comments raised in a previous round of review and you feel that this manuscript is now acceptable for publication, you may indicate that here to bypass the “Comments to the Author” section, enter your conflict of interest statement in the “Confidential to Editor” section, and submit your "Accept" recommendation.

Reviewer #5: (No Response)

Reviewer #6: All comments have been addressed

2. Is the manuscript technically sound, and do the data support the conclusions?

Reviewer #5: Yes

Reviewer #6: Yes

3. Has the statistical analysis been performed appropriately and rigorously? 

Reviewer #5: Yes

Reviewer #6: Yes

4. Have the authors made all data underlying the findings in their manuscript fully available?

Reviewer #5: Yes

Reviewer #6: Yes

5. Is the manuscript presented in an intelligible fashion and written in standard English?

Reviewer #5: Yes

Reviewer #6: Yes

6. Review Comments to the Author

Reviewer #5: The authors, by responding to the reviewers' comments, questions, and remarks, have significantly improved the quality and scholarly sound of the manuscript. I have no further comments. I fully support the publication.

Reviewer #6: The authors have adequately addressed all the comments raised in the previous round of review and have incorporated the suggested revisions. Therefore, I consider the manuscript to be properly revised and ready for publication.

7. PLOS authors have the option to publish the peer review history of their article (what does this mean? ). If published, this will include your full peer review and any attached files.

**Do you want your identity to be public for this peer review?** For information about this choice, including consent withdrawal, please see our Privacy Policy .

Reviewer #5: No

Reviewer #6: No

---

## [Editor Report · Acceptance letter]

PONE-D-24-33400R4

PLOS ONE

Dear Dr. Allen,

I'm pleased to inform you that your manuscript has been deemed suitable for publication in PLOS ONE. Congratulations! Your manuscript is now being handed over to our production team.

Kind regards,

on behalf of

Dr. Faham Khamesipour

Academic Editor

PLOS ONE